# Late to post-Variscan basement segmentation and differential exhumation along the SW Bohemian Massif, Central Europe

Andreas Eberts[1], Hamed Fazlikhani[1], Wolfgang Bauer[1], Harald Stollhofen[1], Helga de Wall[1], Gerald Gabriel[2,3]

[1]GeoZentrum Nordbayern, Friedrich-Alexander-Universität (FAU) Erlangen-Nürnberg, Schlossgarten 5, 91054 Erlangen, Germany.
[2]Leibniz-Institut für Angewandte Geophysik, Stilleweg 2, 30655 Hannover, Germany.
[3]Institut für Geologie, Leibniz Universität Hannover, Callinstraße 30, 30167 Hannover, Germany.

*Correspondence to*: Andreas Eberts (andreas.eberts@outlook.de)

**Abstract**. The exposed Variscan basement in Central Europe is well-known for its complex structural and lithological architecture resulting from multiple deformation phases. We studied the southwestern margin of the Bohemian Massif, which is characterized by major and long-lived shear zones, such as the Pfahl and Danube shear zones, extending over >100 km and initiated during Variscan tectonics. We integrated Bouguer gravity anomaly and LiDAR topographic data analyses and combined our results with available data and observations from low-temperature thermochronology, metamorphic grades, and the exposed granite inventory to detect patterns of basement block segmentation and differential exhumation. Three NW-SE striking basement blocks are bordered by the Runding, Pfahl, and Danube shear zones from the northeast to the southwest. Basement block boundaries are indicated by abrupt changes in measured gravity patterns and metamorphic grades. By applying high-pass filters to gravity data in combination with lineament analysis, we identified a new NNW-SSE striking tectonic structure (Cham Fault), which further segments known basement blocks. Basement blocks that are segmented by the Cham Fault differ in the abundance and spatial distribution of exposed late Variscan granites and are further characterized by variations of apparent thermochronological age data. Based on our observations and analyses, a differential exhumation and tectonic tilt model is proposed to explain the juxtaposition of different crustal levels. Block segmentation along the NW-SE striking Pfahl and Runding shear zones most likely occurred prior, during, and after late orogenic granite emplacement at ca. 320±10 Ma, as some of the granites are cross-cut by the shear zones while others utilized these structures during magma ascent and emplacement. In contrast, activity and block segmentation along the Cham Fault occurred after granite emplacement as the fault sharply truncates the granite inventory. Our study provides evidence for intense and continuous fault activities during late and post-orogenic times and highlights the importance of tectonic structures in the exhumation and juxtaposition of different crustal levels and the creation of complex lithological patterns in orogenic terrains.

# 1    Introduction

The Bohemian Massif extends over ca. 90,000 km² and represents one of the largest coherent exposures of Variscan basement rocks in Western and Central Europe. Originating from the collision of the Laurussia and Gondwana supercontinents during the Paleozoic, the Variscan Orogen formed a belt of low- to high-grade metamorphic and plutonic rocks (summary in Schulmann et al., 2014; Fig. 1a).

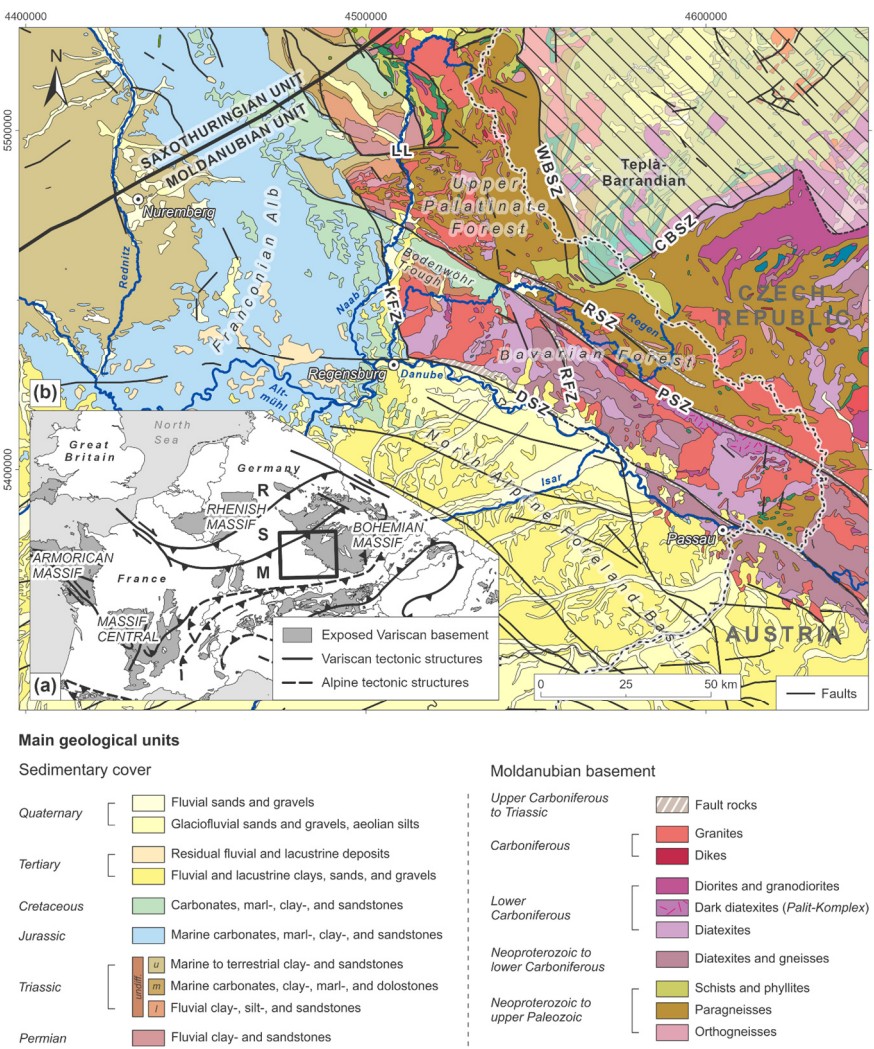

**Figure 1 (a)** Main areas exposing Variscan basement rocks in Central Europe, including traces of major Variscan and Alpine tectonic structures (compiled from Matte, 1986; Franke, 1989; Neubauer and Handler, 2000; Barrier et al., 2004; Asch, 2005). **(b)** Geological and tectonic framework of the southwestern Bohemian Massif (compiled from Freudenberger and Schwerd, 1996; Toloczyki et al., 2006; Teipel et al., 2008; Galadí-Enríquez et al., 2009b). Variscan zones: *R* Rhenohercynian Zone, *S* Saxothuringian Zone, *M* Moldanubian Zone. Fault zones: *CBSZ* Central Bohemia Shear Zone, *DSZ* Danube Shear Zone, *KFZ* Keilberg Fault Zone, *LL* Luhe Line, *PSZ* Pfahl Shear Zone, *RSZ* Runding Shear Zone, *RFZ* Rattenberg Fault Zone, *WBSZ* West Bohemia Shear Zone.

Variscan convergence was followed by widespread strike-slip faulting during the late orogenic evolution between ca. 360 and 320 Ma (e.g., Echtler and Chauvet, 1992; Krohe, 1996; Stephan et al., 2016). As a result, the Bohemian Massif hosts some of the most important fault zones of Central Europe, such as the NW-SE striking Pfahl and Danube shear zones (Fig. 1b). Multiple phases of tectonic activity during the Permian to Paleogene initiated new sets of brittle faults and reactivated pre-existing ductile and brittle structures, creating complex structural patterns (e.g., Horn et al., 1986; Meyer, 1989; Wallbrecher et al., 1991; Kley and Voigt, 2008; Siebel et al., 2010; Schaarschmidt et al., 2019).

The outlined relatively simple succession of tectonic events does not reflect the real complexity of the Bohemian Massif's structure, juxtaposing high-grade metamorphic domains (e.g., the Moldanubian Unit) against low-grade domains (e.g., the Teplá-Barrandian Unit; Krohe, 1996; Cymerman et al., 1997; Kroner et al., 2008). At the local scale, complex lithological and metamorphic patterns can also be observed within each metamorphic unit (Krohe, 1996), which is especially evident in the Moldanubian Unit along the southwestern Bohemian Massif (Fig. 1b). Several studies have contributed to the decipher-ing of the geochronological and geochemical character of that area, especially focusing on the magmatic evolution of the granitic intrusions during the late Variscan tectonothermal event (Finger and Clemens, 1995; Chen et al., 2003; Chen and Siebel, 2004; Dietl et al., 2005; Siebel et al., 2006b; Siebel et al., 2008; Klein et al., 2008; Galadí-Enríquez et al., 2010; Finger et al., 2010). However, the causes for the observed juxtaposition of various metamorphic units and the role of fault reactivation and associated upper crustal vertical movements have not yet been investigated in detail.

In this paper, we apply an integrated methodological approach, combining the analysis of filtered gravity anomaly data, high-resolution Digital Elevation Models (DEMs), and published thermochronological data to reveal the spatial distribution of exposed Variscan units and their boundaries along the southwestern Bohemian Massif. In this context, we also discuss the role of upper crustal fault zones in the exhumation of different crustal levels and the observed juxtaposition of varying litho-logical domains, the latter being one of the most characteristic features of the entire Variscan Orogen (e.g., Krohe, 1996).

## 2 Geological and tectonic setting of the southwestern Bohemian Massif

The late stages of the Variscan Orogeny were characterized by the initiation and reactivation of crustal-scale fault zones, intense HT/LP metamorphic overprint, and extensive crustal melting, which is evidenced by complex patterns of low- to high-grade metamorphic domains, plutonic bodies, and crustal-scale tectonic lineaments (e.g., Franke, 1989, 2000; Echtler and Chauvet, 1992; Krohe, 1996; Büttner, 2007; Kroner and Romer, 2013; Schulmann et al., 2014; Stephan et al., 2016). Along the southwestern Bohemian Massif, this late-stage evolution of the Variscan Orogeny is manifested by the presence of significant fault zones, such as the Pfahl and Danube shear zones, and vast migmatite complexes intruded by voluminous late orogenic granite bodies (Wallbrecher et al., 1991; Brandmayr et al., 1995; Finger and Clemens, 1995; Kalt et al., 1999; Kalt et al., 2000; Propach et al., 2000; Chen et al., 2003; Siebel et al., 2003; Klein et al., 2008). Two distinct tectonometamorphic units bounded by large fault zones are (1) the Moldanubian (*sensu stricto,* further referred to as the Moldanubian Unit) and (2) the Teplá-Barrandian (Kossmat, 1927; Franke, 1989; Fig. 1b). The Moldanubian Unit is comprised of high-grade meta-

morphic rocks predominantly formed during the Variscan HT/LP tectonothermal events (e.g., Grauert et al., 1974; Franke, 1989). The Teplá-Barrandian Unit is considered a supracrustal unit consisting of a low-grade Neoproterozoic basement (e.g., Franke, 1989; Žák et al., 2014). In this study, we focus on basement segmentation and exhumation of the Moldanubian Unit exposed at the southwestern margin of the Bohemian Massif.

## 2.1 Origin and characteristics of the Moldanubian basement units

During the ceasing Cadomian Orogeny in the late Neoproterozoic/early Paleozoic, the area of the nowadays southwestern Bohemian Massif was situated along the northern margin of Gondwana (e.g., Rohrmüller et al., 1996; Linnemann et al., 2004; Fatka and Mergl, 2009; Žák and Sláma, 2018). A succession of mainly pelitic greywackes, intercalated with igneous rocks, are thereby considered as the protoliths of the so-called "Monotonous Group". The latter is characterized by a series of biotite-plagioclase-bearing para- and orthogneisses with only a minor abundance of quartzites, marbles, graphitic schists, amphibolites, and granitic gneisses (Rohrmüller et al., 1996; Franke, 2000; Kroner et al., 2008). On the other hand, volcano-sedimentary successions, probably formed in response to continental rifting and associated volcanism, are documented within the so-called "Varied Group". This group is characterized by paragneisses with higher abundances of quartzites, marbles, graphitic schists, and amphibolites (Rohrmüller et al., 1996; Kroner et al., 2008). In the southeastern part of the Bavarian Forest, the diatexites and their enclaves are interpreted as derivatives of a series of dacites, andesites, and basalts, which suggests the presence of an ancient island arc (Propach et al., 2008).

Late Neoproterozoic/early Paleozoic units of the southwestern Bohemian Massif were overprinted by HT/LP metamorphism during the late stages of the Variscan Orogeny. Pressures of up to 7 kbar (i.e., ca. 20 km burial depth) and temperatures of >800 °C led to partial melting and the formation of vast migmatite complexes (Grauert et al., 1974; Kalt et al., 1999; Kalt et al., 2000), the anatectic grade of which is increasing from NNE to SSW (Blümel, 1972; Teipel et al., 2008; Galadí-Enríquez et al., 2009b). Whereas this metamorphic event has been dated to ca. 335 to 340 Ma in the southern central Bohemian Massif, younger thermal overprints and partial anatexis are documented for the Bavarian part of the Moldanubian Unit, with metamorphic ages progressively decreasing towards the southwest (ca. 320 to 315 Ma in the area to the southwest of the Pfahl Shear Zone, Kalt et al., 2000; Propach et al., 2000; Gerdes et al., 2006; Finger et al., 2007; Siebel et al., 2012; compiled by Teipel et al., 2008). Contemporaneously, voluminous granite bodies intruded large parts of the southwestern Bohemian Massif (Table 1). Estimates of granite emplacement depths in the study area are restricted to the southeastern Bavarian Forest (Fig. 1b and Table 1). They vary between 14-15 km for the Saldenburg Granite, which is part of the Fürstenstein Composite Massif (Dietl et al., 2005), and 16-18 km for the Hauzenberg Granite II, which is part of the Hauzenberg Composite Massif (Klein et al., 2008). Erosional products of these late orogenic granites were deposited in the adjacent Carboniferous-Permian basin (Naab Trough *sensu* Schröder, 1988), indicating their rapid exhumation shortly after their late Carboniferous emplacement (Welzel, 1991; Mielke, 1993; Galadí-Enríquez et al., 2009a).

**Table 1** Ages and estimated emplacement depths for granites along the southwestern Bohemian Massif. All ages have been measured on zircons. For granite locations, see Fig. 3. [a]Klein et al. (2008), [b]Chen and Siebel (2004), [c]Dietl et al. (2005), [d]Siebel et al. (2008), [e]Siebel et al. (2006a), [f]Siebel et al. (2010), [g]Chen et al. (2003). *ND* not determined.

| Name | Variety | Age (Ma), method | Est. emplacement depth (km) |
|---|---|---|---|
| Hauzenberg Composite Massif | *Hauzenberg Granite II* | 320±3[a], U-Pb | 16-18[a] |
| Fürstenstein Composite Massif | *Tittling Granite* | 322-324[b], Pb-Pb | ND |
| | *Eberhardsreuth Granite* | 312-316[b], Pb-Pb | ND |
| | *Saldenburg Granite* | 312-318[b], Pb-Pb | 14-15[c] |
| Haidel Massif | | 323±4[d], Pb-Pb | ND |
| Rinchnach Stock | | 320-329[e], Pb-Pb | ND |
| Patersdorf Stock | | 322-323[e], Pb-Pb | ND |
| Metten Massif | | 324±5[d], Pb-Pb | ND |
| Arnbruck Stock | | 325±2[d], Pb-Pb | ND |
| Regensburg Forest Massif | *Kristallgranit* | 325±7[f], Pb-Pb | ND |
| Neunburg Massif | | 319±4[g], U-Pb | ND |
| Oberviechtach Stock | | ca. 320[g], U-Pb | ND |

## 2.2 Structural characteristics

The tectonic configuration of the southwestern Bohemian Massif during the late stages of the Variscan Orogeny is interpreted as a conjugate shear system related to N to NNW directed shortening (Wallbrecher et al., 1991; Brandmayr et al., 1995; Peterek et al., 1996; Büttner, 1999; Galadí-Enríquez et al., 2010). The NW-SE trending Pfahl Shear Zone strikes over 150 km from northern Austria to southeast Germany; it is the best-known and one of the most prominent examples among the resulting dextral strike-slip faults (e.g., Brandmayr et al., 1995; Table 2). The Pfahl Shear Zone marks a pronounced change in the anatectic grade, separating the metatectic-dominated domain of the northeastern Bavarian and Upper Palatinate forests from the diatectic-dominated domain of the southwestern Bavarian Forest (Teipel et al., 2008; Galadí-Enríquez et al., 2009b; Fig. 1b). Changes in anatectic grades and variations of the granite geochemistry across the Pfahl Shear Zone are interpreted as reflecting differential exhumation with deeper crustal levels exposed to the southwest of the shear zone (Grauert et al., 1974; Beer, 1981; Finger and Clemens, 1995; Siebel et al., 2008; Finger and Rene, 2009). In addition, analysis of quartz mineralizations along the Pfahl Shear Zone gives an indication for the exposure of deeper crustal levels along the southeastern segment of the shear zone (Schaarschmidt et al., 2019). Two contrasting interpretations of the origin of the Pfahl Shear Zone exist, i.e., (I) the shear zone represents a former suture zone along which two different basement terranes amalgamated (Siebel et al., 2008; Siebel et al., 2009) or (II) the shear zone just intersects the otherwise continuous Moldanubian Unit (Finger et al., 2007; Finger and Rene, 2009; Finger et al., 2010).

Subparallel and ca. 10 km north of the Pfahl Shear Zone, the Runding Shear Zone represents another important tectonic lineament in the study area (Fig. 1b, Table 2). Similar to the Pfahl Shear Zone, the Runding Shear Zone marks a pronounced change in anatectic grades, separating the metatectic-dominated domain of the northeastern Bavarian Forest from higher-grade diatectic rocks in between the Pfahl and Runding shear zones (Teipel et al., 2008; c.f., Fig. 3).

In the southwest, the exposed Moldanubian basement is delimited by the Danube Shear Zone (Fig. 1b). Similar to the Pfahl Shear Zone, the Danube Shear Zone is characterized by multiple deformation phases lasting from the late Paleozoic until the Cenozoic (Table 2). The structural patterns in between these two major shear zones are interpreted as a consequence of a dextral Riedel-type shear system that developed in response to a "rift-and-wrench" tectonic phase under a NNW-SSE to N-S directed compressional stress field (Zeitlhöfler, 2007).

**Table 2** Main inferred tectonic activity phases, kinematics, and estimated minimum displacements (Est. displ.) of the six major fault zones in the Moldanubian Unit of the southwestern Bohemian Massif. [a]Mattern (1995), [b]Siebel et al. (2010), [c]Freudenberger (1996), [d]Führer (1978), [e]Bauberger and Cramer (1961), [f]Carlé (1955), [g]Seemann (1925), [h]Meyer (1989), [i]Müller (1994), [j]Peterek et al. (1996), [k]Siebel et al. (2005), [l]Galadí-Enríquez et al. (2010), [m]Horn et al. (1986), [n]Meyer (1993), [o]Rohrmüller et al. (2017), [p]Zeitlhöfler (2007). *ND* not determined.

| Name | Strike | Inferred activity phase | Main kinematics | Est. displ. (km) |
|---|---|---|---|---|
| Danube Shear Zone | NW-SE | Carboniferous[a,b] - Permian[b] | Dextral strike-slip[a,b] | ND |
| | | Mesozoic - Cenozoic[b,c] | Normal[d] | 1.3[c] |
| Keilberg Fault Zone | NNW-SSE | Late Paleozoic - Paleogene[e,f,g] | Reverse[c], Normal[h] | 1.2[c] |
| Luhe Line | E-W | Carboniferous - Permian[i] | Reverse[i] | 2[i] |
| | | Early Cretaceous[j] | Normal[j] | ND |
| | | Late Cretaceous - Paleogene[j] | Reverse[j] | ND |
| Pfahl Shear Zone | NW-SE | Carboniferous[k,l] | Dextral strike-slip[a,l] | ND |
| | | Permian[m] | ND | ND |
| | | Late Cretaceous - Paleogene[n,h] | Reverse[n,h] | 0.5[n] |
| Rattenberg Fault Zone | NNW-SSE | Late Paleozoic[o] | ND | ND |
| Runding Shear Zone | NW-SE | Carboniferous - Permian[p] | Dextral strike-slip[o] | ND |

## 3 Data and methodologies

This study uses an integrated approach combining the analysis of Bouguer gravity anomaly data and Digital Elevation Models (DEMs) to unravel the lithological and structural architecture of the exposed Variscan crust in the southwestern Bohemian Massif.

### 3.1 Gravity analysis

The Bouguer gravity dataset is part of a pool of ca. 350,000 reprocessed and merged gravity data points in Germany and surrounding areas with a mean point spacing of 2-3 km and an overall accuracy of ± 100 µGal (Leibniz-Institut für Angewandte Geophysik, 2010; Skiba, 2011). In this study, we compare surface geology with unfiltered and filtered Bouguer gravity anomaly data to identify potential granites in the subsurface and to reveal the spatial relationship to their exposed counterparts. We applied 20 and 30 km wavelength high-pass filters to the Bouguer gravity anomaly data (using Oasis Montaj, Seequent) to identify regional and local anomaly sources in the subsurface. The 20 km high-pass filter mainly includes the gravity signature of causative bodies located in the shallower subsurface, while a 30 km high-pass filter also considers somewhat deeper crustal bodies (e.g., Lowrie, 2007). In areas with exposed crystalline rocks, local circular and semi-circular negative gravity anomalies are often related to exposed and buried granites due to the density contrast between granites and the higher-density metamorphic country rocks (e.g., Behr et al., 1989; Trzebski et al., 1997; Siebel et al., 1997; Sedlák et al., 2007; Sedlák et al., 2009). To confirm the density difference between granites and surrounding metamorphic rocks, we collected density data from exposed granites and compared these to published metamorphic rock densities. Granite densities were measured by applying a buoyancy technique in isopropanol (Archimedes' principle). Samples were either collected as loose rocks or, if applicable, as drilled cores of two to five centimeters diameter to ensure "fresh" samples without major cracks.

### 3.2 Topographic analysis

The DEM used in this study is based on LiDAR (Light Detection and Ranging) point clouds acquired by the Bavarian Agency for Digitisation, High-Speed Internet and Surveying. Only the last returned laser pulses have been considered in the dataset so that the final DEM represents the landscape's elevation without being affected by the vegetational cover. Each data point has been georeferenced, referred to the Gauss-Krüger Coordinate System (zone 4) and the German Combined Quasigeoid 2011 (GCG2011) for precise horizontal and vertical positioning, achieving horizontal accuracies of ± 0.5 m and vertical accuracies of ± 0.2 m. DEM data were resampled in ArcGIS Pro (ESRI) to a spatial resolution of 10 m, which ensures efficient data processing with sufficient detail to perform lineament mapping and analysis in the study area.

#### 3.2.1 Topographic swath profiles

Traditional elevation profiles along single lines can only image a small fraction of all topographic characteristics, which is related to their small footprint. In contrast, topographic swath profiles are able to illustrate even complex landscapes without neglecting the spatial distribution of important but spatially limited morphological features such as local peaks and troughs (e.g., Telbisz et al., 2013). We used topographic swath profiles to illustrate the overall topographic expression of the study area. Swath profiles are constructed by projecting the elevation data within a rectangular swath onto its longitudinal axis.

Depending on the spatial extent of the studied area, the swath can have widths ranging from 100s of meters to 10s of kilometers.

Swath profile data were obtained using the ArcGIS Add-in "SwathProfiler" (Pérez-Peña et al., 2017). We used a fixed swath width of 10 km for all of the profiles. This value is large enough to summarize the present topographic variations adequately while small enough to avoid mixing of morphologies from very different landscapes. Elevation data were sampled along 50 parallel profiles with a step size of 15 m (i.e., 1.5 times the DEM resolution). Important statistical parameters such as the maximum, minimum, and mean elevation can be illustrated simultaneously, providing information about both the general spatial distribution of elevations within an area but also of discrete topographic features such as local peaks and troughs, paleo-surfaces, and incised valleys.

### 3.2.2 Lineament analysis

The analysis of topographic lineaments yields valuable information on the structural patterns of large areas, especially if remote or inaccessible. In cases where active tectonic processes prevail over erosional processes and leveling of the landscape by sediment mobilization and deposition, fault slip is often immediately transferred to the Earth's surface, forming distinct topographic lineaments (i.e., fault scarps) that can be easily traced and interpreted using aerial photographs, DEMs, or field mapping (e.g., Stewart and Hancock, 1990; Keller and Pinter, 2002; Burbank and Anderson, 2012). If the time between fault rupture is long or tectonic activity ceases, however, the scarp will degrade and geomorphic processes will modify the hillslopes and channels of the landscape towards a new equilibrium. In such a case, tectonic structures will no longer be visible as well-defined fault scarps but, because they often induce gradients in rock erodibility, as linear to curvilinear river valleys, ridgelines, or slope breaks (e.g., Jordan et al., 2005; Fürst et al., 1978; Fig. 2). Strike-slip faults thereby tend to form symmetrical morphologies such as river valleys or ridgelines (e.g., Fürst et al., 1978; Fig. 2). High-angle normal and reverse faults typically form linear slope breaks (Fig. 2), whereas low-angle thrust faults tend to appear somewhat irregular in topography (Fürst et al., 1978; Drury, 1987; Prost, 1994; Goldsworthy and Jackson, 2000).

Lineaments were detected using DEM derivatives such as hillshade, slope, curvature, and aspect maps as enhancement tools to identify linear features in topography. A shortcoming of traditional hillshade maps is that they induce directional biases due to their heterogeneous way of illuminating the landscape, which is mainly a result of the fixed azimuth of the artificial light source (e.g., Scheiber et al., 2015). To reduce this bias, we used multi-directional hillshades to map topographic lineaments, illuminating landscapes in a more homogeneous way.

Calculations of DEM derivatives and the mapping procedure were carried out in ArcGIS Pro (ESRI). Statistical analysis was accomplished by using a modified routine in MATLAB (MathWorks) that partly relies on the FracPaQ toolbox (Healy et al., 2017).

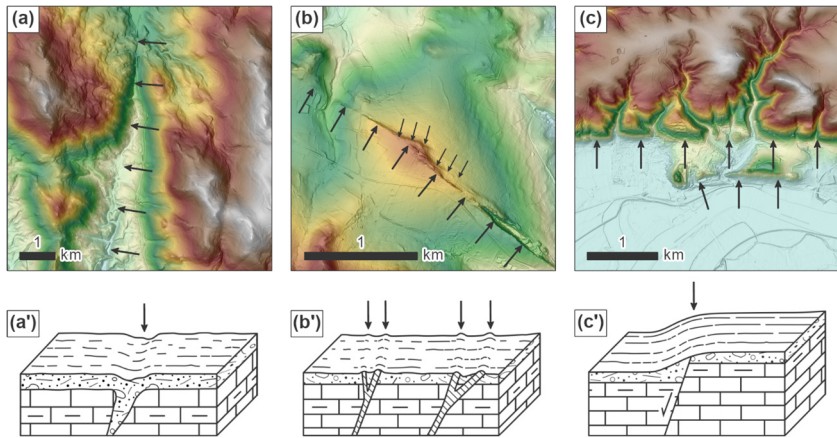

**Figure 2** Map views (**a**) to (**c**) and sketches (**a'**) to (**c'**) illustrating the main lineament types encountered along the southwestern Bohemian Massif (sketches redrawn and modified from Fürst et al., 1978). Arrows depict the traces of lineaments. (**a**)-(**a'**) River valleys tend to form along zones of weakened bedrock commonly induced by faults and fractures. The shown example depicts the valley of the river Kollbach located in the Bavarian Forest. (**b**)-(**b'**) Linear ridgelines may form in cases where minerals, which are more resistant to erosion compared to the adjacent country rocks, have been precipitated within an open fault or fracture system. The most well-known example of this lineament type in the study area is the quartz lode of the Pfahl Shear Zone. (**c**)-(**c'**) Vertical movements along normal and reverse faults are usually expressed as abrupt, linear slope breaks. The shown example is located along the northwestern segment of the Danube Shear Zone.

## 4 Basement lithological configuration of the southwestern Bohemian Massif and its relation to the structural architecture

Three main basement domains are defined in the study area based on their predominant metamorphic grades (Fig. 3). Basement domains A, B, and C are bounded by the essentially NW-SE striking Pfahl, Danube, and Runding shear zones (Fig. 3b). By considering the spatial distribution of exposed late Variscan granites, defined basement domains are further subdivided into (I) domains A1, B, and C1, where late Variscan granites are less exposed and are preferentially aligned with NW-SE striking shear zones, and (II) domains A2 and C2, where late Variscan granites are abundant and not necessarily aligned with major shear zones (Fig. 3b).

### 4.1 Domain A: Diatexite-dominated

Domain A exposes rocks that have experienced an advanced stage of anatexis, with diatexites forming the predominant rock type (ca. 45 % at the present erosional level, Fig. 3a). In the central part, gneissic rocks are intercalated with diatexites (Fig. 3b). Lower-grade paragneisses are restricted to the so-called "Donauleiten-Serie" in the very southeast of domain A (Daurer, 1976; Fig. 3b). Late Variscan granites in domain A are extensively exposed in the northwest, whereas in the central part, a very limited number of granitic bodies is exposed, which are restricted to the traces of the Pfahl and Danube shear zones (e.g., the Metten Massif and the Patersdorf Stock, Fig. 3b). Based on the abundance and distribution of exposed late Variscan granites, a distinct boundary subdivides domain A into domain A2, i.e., the northwestern part with abundant granites, and

A1, i.e., the central-southeastern part, where granites are solely exposed along the traces of major fault zones (Fig. 3b). This boundary also defines the southeastern border of the Mesozoic Bodenwöhr Trough and marks the western limit of the ca. 10 km wide, NNW-SSE striking Stallwang Fault Zone, implying a tectonic origin of the boundary (Troll, 1967). Towards the very southeast of domain A1, the abundance of granites increases again and their exposure is not aligned with the Pfahl and Danube shear zones (Fig. 3b). Here, the diatexites and their enclaves are interpreted as derivatives of a series of dacites, andesites, and basalts, suggesting the presence of an ancient island arc (Propach et al., 2008). Despite the different lithological character of the southeastern part of domain A1 (i.e., ortho-anatexites and extensively exposed granites), a distinct boundary separating this part from the northwestern part of domain A1 (i.e., para-anatexites and fault-related granites) is not observed. Hence, we interpret the southeastern part of domain A1 as a lithologically different portion of the same (fault-bounded) domain.

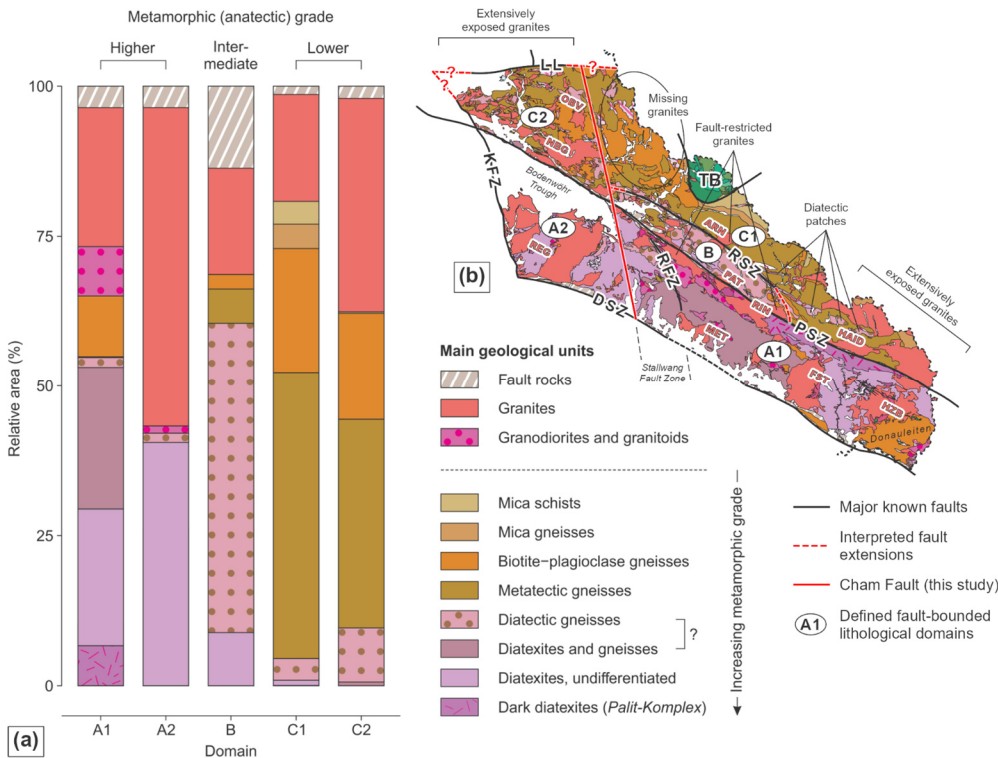

Figure 3 Statistical and spatial distribution of exposed lithologies. Five fault-bounded domains with characteristic rock inventories can be identified (A1, A2, B, C1, and C2). (a) Stacked bar plots illustrating the relative area proportions of the exposed lithologies in each of the domains (calculations based on geological maps from Teipel et al., 2008 and Galadí-Enríquez et al., 2009b). (b) Detailed geological map of the studied area along the southwestern Bohemian Massif (modified from Teipel et al., 2008; Galadí-Enríquez et al., 2009b). An important fault (Cham Fault) is proposed in the northwestern part of the study area separating domains A1 and C1 from A2 and C2. Fault zones: *DSZ* Danube Shear Zone, *KFZ* Keilberg Fault Zone, *LL* Luhe Line, *PSZ* Pfahl Shear Zone, *RSZ* Runding Shear Zone, *RFZ* Rattenberg Fault Zone. Exposed granite bodies: *ARN* Arnbruck Stock, *FST* Fürstenstein Composite Massif, *HAID* Haidel Massif, *HZB* Hauzenberg Composite Massif, *MET* Metten Massif, *NBG* Neunburg Massif, *OBV* Oberviechtach Stock, *PAT* Patersdorf Stock, *REG* Regensburg Forest Massif, *RIN* Rinchnach Stock; names after Klominský et al. (2010). *TB* Teplá-Barrandian.

## 4.2 Domain B: Diatectic gneiss-dominated

Domain B is bounded by the Pfahl and Runding shear zones and is characterized by the exposure of predominantly diatectic gneisses (ca. 50 % of the total area) and granites (ca. 20 %, Fig. 3a). Diatectic gneisses are almost entirely restricted to this domain. Outside domain B, small patches of diatectic gneisses are aligned subparallel to the Pfahl Shear Zone in the southeast of domain C (Fig. 3b). Late Variscan granites in domain B are mainly aligned with the Pfahl and Runding shear zones (Fig. 3b). A high amount of fault rocks, predominantly mylonites and cataclasites (ca. 15 % of the total area), indicate pervasive faulting in domain B.

## 4.3 Domain C: Metatectic gneiss-dominated

Domain C comprises the area to the northeast of the Pfahl and Runding shear zones and is characterized by the presence of intermediate- to high-grade metamorphic rocks. The metamorphic grades in domain C are lower compared to domains A and B (Fig. 3). Tectonostratigraphic units in domain C consist of intermediate-grade biotite-garnet-chlorite schists, higher-grade mica- and biotite-plagioclase gneisses, and metatectic cordierite-sillimanite-K-feldspar gneisses, the latter representing ca. 45 % of the total area of domain C (Fig. 3a). Exposed late Variscan granites in the northwest of domain C (i.e., domain C2) terminate towards the center at the NNW extension of the previously defined boundary that subdivides domain A into A1 and A2 (Fig. 3b). Due to its pervasiveness, we interpret this boundary to represent an important structural feature in the study area and call it "Cham Fault". East of the Cham Fault, in the central part of domain C, granites are solely exposed along the Runding Shear Zone (e.g., the Arnbruck Stock, Fig. 3b). Towards the southeast of domain C, exposure of granites gradually increases, thus showing a similar pattern as observed in domain A.

## 5 Subsurface distribution of late Variscan granites

The spatial distribution of exposed late Variscan granites with a sharp decrease in abundance across the Cham Fault raises the question whether the granites are generally not present in the central parts of the study area or whether they have not yet been exhumed to the surface. Comparing the unfiltered and high-pass filtered Bouguer gravity data with the geological maps indicates that known granites are associated with circular to semi-circular gravity lows (e.g., Regensburg Forest Massif, Patersdorf Stock, Fürstenstein Composite Massif, Hauzenberg Composite Massif, Fig. 4). The densities of exposed granites in the study area range from 2640 to 2680 kg/m³, whereas the metamorphic country rocks show densities >2690 kg/m³ (Table 3). Distinct local and circular to semi-circular gravity lows along the western Bohemian Massif that are not associated with exposed granites are generally interpreted as buried granite bodies (e.g., Behr et al., 1989; Trzebski et al., 1997; Sedlák et al., 2009; de Wall et al., 2019). Applying 20 and 30 km high-pass filters confirms the presence of granitic bodies (circular and semi-circular gravity lows) at different crustal levels (Fig. 4c,d). Some of these subsurface granites trend subparallel to major fault zones, similar to their exposed counterparts (Fig. 4). The presence and distribution of buried granites, as evi-

denced by filtered gravity anomaly maps, suggest a rather homogenous distribution of granites in the subsurface of the study area (Fig. 4c,d).

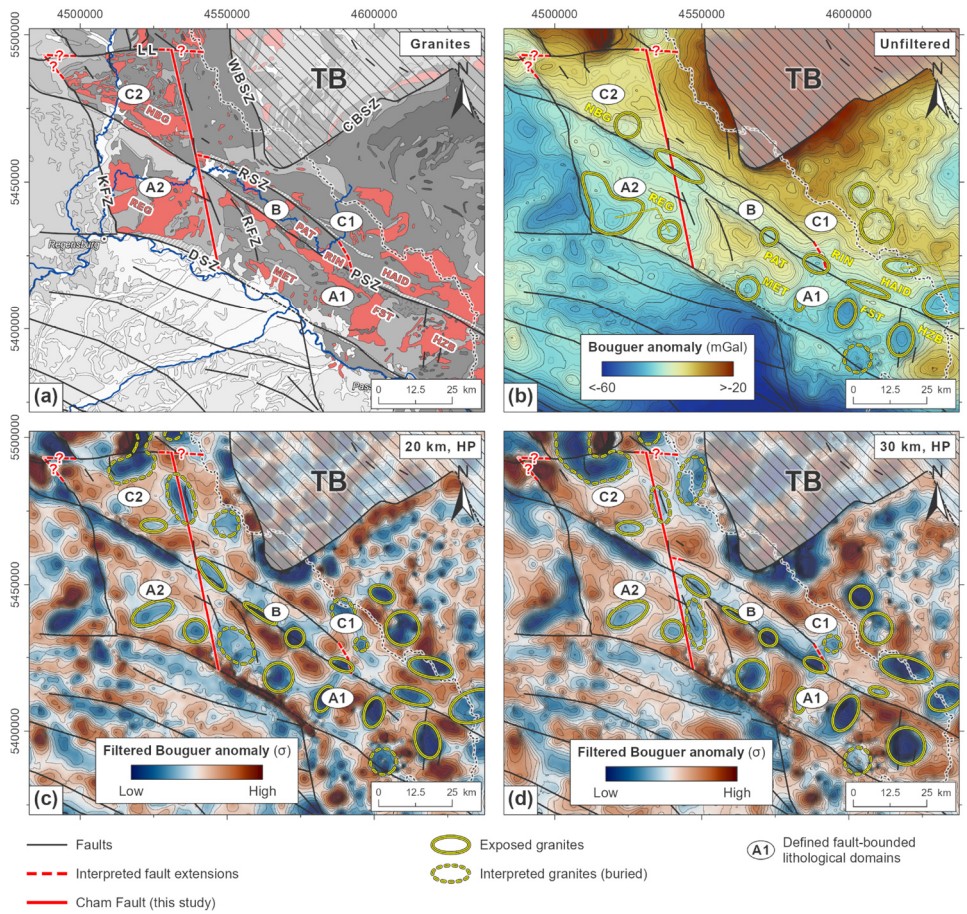

**Figure 4** Compilation of unfiltered and filtered gravity data. (**a**) Gray-scaled geological map of the southwestern Bohemian Massif with exposed late Variscan granites highlighted in red (modified from Freudenberger and Schwerd, 1996; Toloczyki et al., 2006; Teipel et al., 2008; Galadí-Enríquez et al., 2009b). (**b**) Unfiltered Bouguer anomaly map of the study area and its surroundings (data source: Leibniz-Institut für Angewandte Geophysik, 2010; Skiba, 2011). (**c**)-(**d**) High-pass filtered gravity data (20 km and 30 km wavelength, respectively) depicting the upper crustal configuration of the study area. The data are color-coded with respect to their standard deviation (σ) to highlight relative gravity highs and lows. Traces of major fault zones and the interpreted Cham Fault, as well as the locations of exposed and interpreted granite bodies, are shown. The scientific color maps "roma" (b) and "vik" (c,d) (Crameri, 2021) are used to prevent visual distortion of the data and exclusion of readers with color-vision deficiencies (Crameri et al., 2020). Fault zones: *CBSZ* Central Bohemia Shear Zone, *DSZ* Danube Shear Zone, *KFZ* Keilberg Fault Zone, *LL* Luhe Line, *PSZ* Pfahl Shear Zone, *RSZ* Runding Shear Zone, *RFZ* Rattenberg Fault Zone, *WBSZ* West Bohemia Shear Zone. Exposed granite bodies: *FST* Fürstenstein Composite Massif, *HAID* Haidel Massif, *HZB* Hauzenberg Composite Massif, *MET* Metten Massif, *NBG* Neunburg Massif, *PAT* Patersdorf Stock, *REG* Regensburg Forest Massif, *RIN* Rinchnach Stock; names after Klominský et al. (2010). *TB* Teplá-Barrandian.

The metamorphic rocks surrounding the granites show different gravity signatures (Schaarschmidt et al., 2019). Areas dominated by gneissic rocks generally show less pronounced negative Bouguer anomalies (domain C, Fig. 4b-d). In contrast, rocks of higher metamorphic grades (i.e., diatexites) show more pronounced negative anomalies (domain A, Fig. 4b-d). This relationship can be explained by the density contrasts of the present rock types, with higher-grade metamorphic rocks tending to be less dense compared to lower-grade rocks (Table 3).

**Table 3** Mean densities of granites and metamorphic rocks in the study area. Equivalent data on metamorphic rocks outside the study area are included. [a]own measurements, [b]Guy et al. (2011), [c]Smithson (1971).

| Data type | Rock type | Granite pluton | Mean density (kg/m³) |
|---|---|---|---|
| Study area | Granites | Hauzenberg Composite Massif | 2650±10[a] |
| | | Fürstenstein Composite Massif | 2680±50[a] |
| | | Patersdorf Stock | 2670±10[a] |
| | | Metten Massif | 2670±10[a] |
| | | Regensburg Forest Massif | 2640±20[a] |
| | Metamorphic rocks (approx.) | | 2690-2760[b] |
| Global data | Migmatites | | 2730±60[c] |
| | Biotite gneiss and schist | | 2750±60[c] |
| | Mica schist | | 2800±30[c] |
| | Biotite-hornblende gneiss | | 2860±60[c] |
| | Amphibolite | | 3030±70[c] |

## 6 Topographic analysis

In this chapter, we present the morphological analysis results using swath profiles and the mapping of topographic lineaments to confirm and characterize basement domains identified previously based on the spatial distribution of metamorphic rocks and late Variscan granites.

### 6.1 Across- and along-strike variations in topography

Two swath profiles illustrate across- and along-strike topographic variations of the southwestern Bohemian Massif (Fig. 5). A mountainous landscape marks the northwestern part of domain A1 with peaks up to ca. 1000 m a.s.l. (Fig. 5a,b). This elevated area shows an increased abundance of gneissic rocks (Fig. 5b). In contrast, the southeastern parts of domain A1, which is dominated by higher-grade diatexites, show a low-lying landscape with a reduced relief and elevations down to ca. 350 m a.s.l..

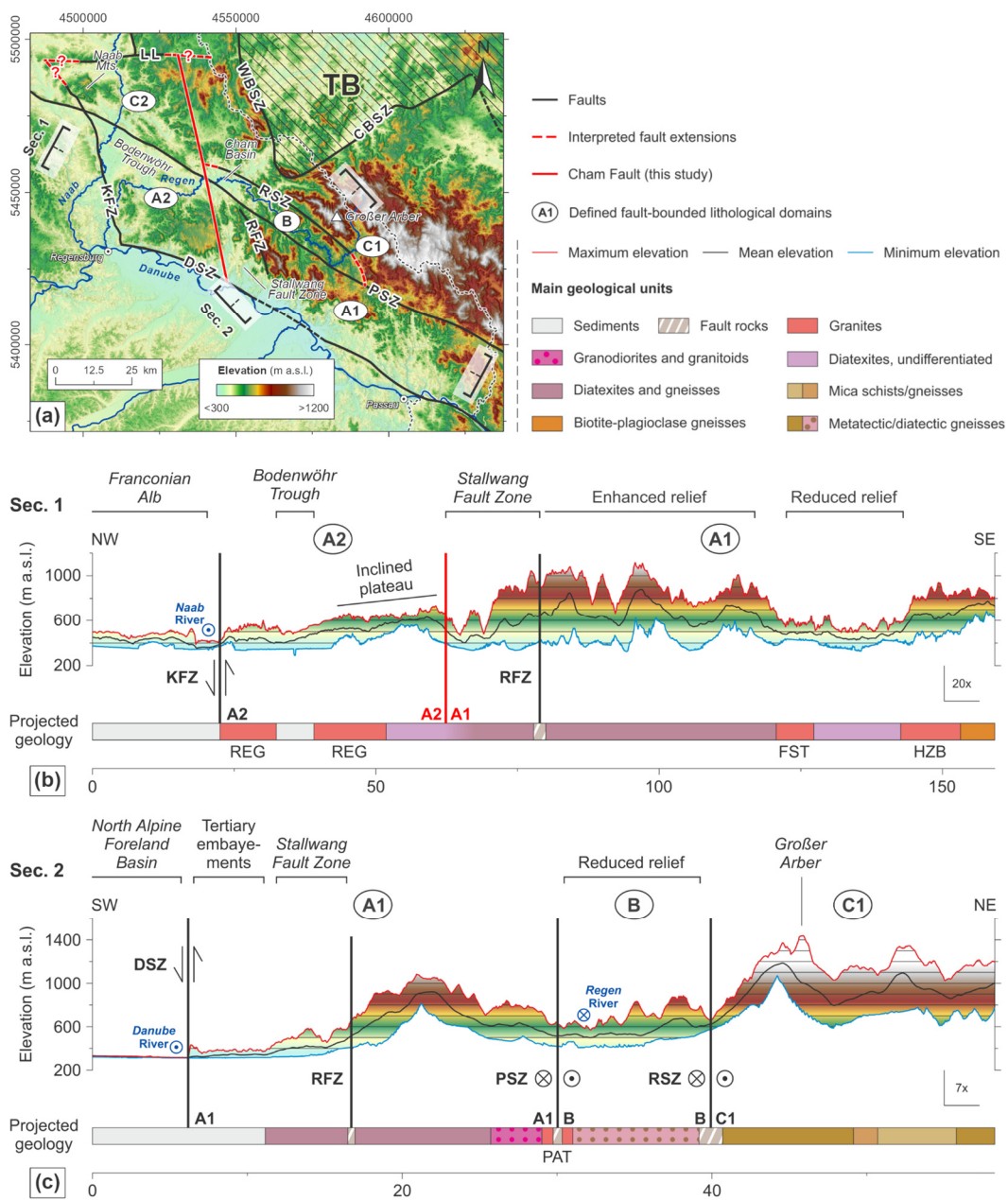

**Figure 5** (**a**) DEM depicting the locations of major tectonic structures and swath profiles. The DEM is based on Shuttle Radar Topographic Mission (SRTM) data with a spatial resolution of ca. 30 m in the study area (Earth Resources Observation And Science Center, 2017). (**b**) Swath profile transecting the study area in a NW-SE direction. (**c**) Swath profile transecting the study area in a SW-NE direction. Important morphological and tectonic features are indicated. For locations of the profiles, see (**a**). Fault zones: *CBSZ* Central Bohemia Shear Zone, *DSZ* Danube Shear Zone, *KFZ* Keilberg Fault Zone, *LL* Luhe Line, *PSZ* Pfahl Shear Zone, *RSZ* Runding Shear Zone, *RFZ* Rattenberg Fault Zone, *WBSZ* West Bohemia Shear Zone. Exposed granite bodies: *FST* Fürstenstein Composite Massif, *HZB* Hauzenberg Composite Massif, *PAT* Patersdorf Stock, *REG* Regensburg Forest Massif; names after Klominský et al. (2010). *TB* Teplá-Barrandian.

In domain A2, the Variscan basement is predominantly exposed to the south of the Regen River and is characterized by an undulating relief with elevations in between ca. 400 and 750 m a.s.l. (Fig. 5a). The very southeast of that area is marked by a distinctly elevated plateau that gently dips towards the northwest (Figs. 5a,b). To the north of the Regen River, the Mesozoic to Cenozoic sediments of the Bodenwöhr Trough evoke a flat relief (Fig. 5a). The Stallwang Fault Zone, which is defined as the area in between the Cham Fault and the Rattenberg Fault Zone, borders domain A2 to the east and is characterized by subparallel, NNW-SSE-trending erosional furrows indicating a highly deformed area that favors erosion (Figs. 5a,b). Domain B, with its high-grade metamorphic rocks (diatectic gneisses), shows lower elevations compared to adjacent areas and catches the drainages of its surroundings (Figs. 5a and 5c).

A very diverse topography characterizes the metatectic-dominated area of domain C1, with its eastern part comprising a high-relief mountainous landscape (Figs. 5a and c). In contrast, in the northwest, a heterogeneous landscape with basins (e.g., the Rötz Basin) that are surrounded by peaks up to 900 m a.s.l. high is developed (c.f. Figs. 6a and 7). Domain C2, which is also dominated by metatectic rocks but comprises a much higher number of exposed granites compared to C1, is characterized by low to intermediate elevations of ca. 350 to 750 m a.s.l.. In its center, the landscape is dissected by the broad Naab River valley, draining to the south (Fig. 5a). To the west of the Naab River, the Naab Mountains form a local fault-bounded topographic high (Fig. 5a).

## 6.2 Topographic lineament analysis

In total, 1315 topographic lineaments have been identified in the study area (Fig. 6). Linear river valleys count for ca. 50 %, ridgelines count for ca. 30 %, and sudden breaks in slope count for ca. 10 % of the identified lineaments. The remaining ca. 10 % of the identified lineaments are related to a combination of valleys, ridgelines, and/or slope breaks. Known faults are thereby often associated with distinct lineaments, suggesting a close relationship between tectonic and topographic features.

### 6.2.1 Topographic signatures of major fault zones

Major fault zones are all well-visible in topography (Fig. 6a). The Pfahl Shear Zone is characterized by two different topographic expressions. Along its northwestern and southeastern segments, the shear zone is expressed by pronounced slope breaks (Fig. 6a and profiles 1 and 3 in Fig. 6b). In the northwest, the Pfahl Shear Zone defines the northern border of the Bodenwöhr Trough, whereas its southeastern segment separates a mountainous landscape in the north from a moderate-relief landscape in the south (Fig. 6a). Along its central segment, the Pfahl Shear Zone forms a narrow but distinct morphological ridge (Fig. 6b, profile 2). This morphological ridge is attributed to the pervasive quartz mineralization and resulting weathering resistance along this segment of the Pfahl Shear Zone (Priehäusser, 1961; Lehrberger et al., 2003).

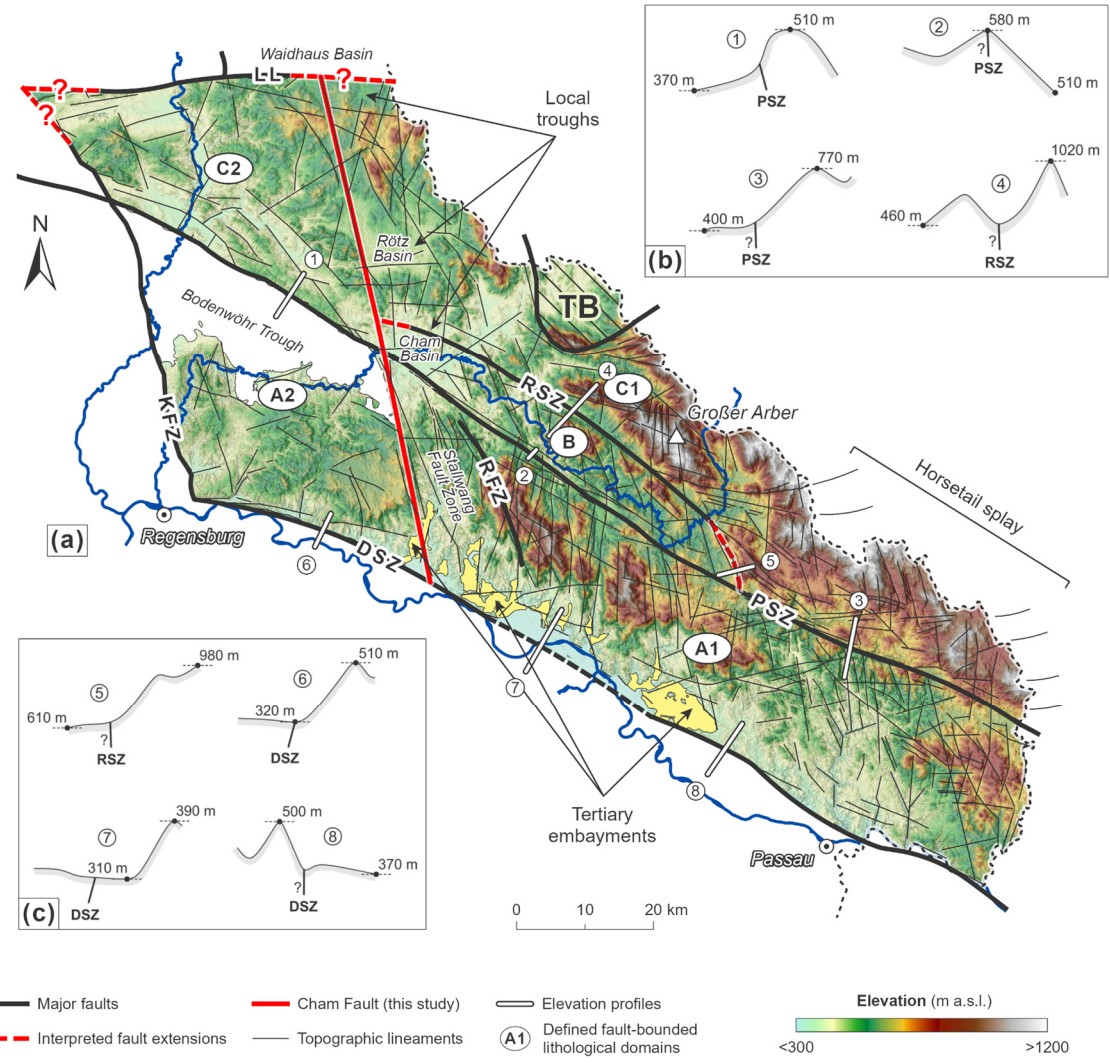

**Figure 6** Lineament inventory of the study area. (**a**) DEM depicting lineaments traces, major fault zones, and the interpreted Cham Fault. The locations of important topographic features like troughs and Tertiary embayments are outlined. Note that only higher-order lineaments are shown to maintain clarity. (**b**)-(**c**) Elevation profiles across the different topographic segments of the Pfahl, Runding, and Danube shear zones (viewing direction from southeast to northwest). Question marks indicate unknown dip-directions of the associated faults. Fault zones: *DSZ* Danube Shear Zone, *KFZ* Keilberg Fault Zone, *LL* Luhe Line, *PSZ* Pfahl Shear Zone, *RSZ* Runding Shear Zone, *RFZ* Rattenberg Fault Zone. *TB* Teplá-Barrandian.

The Runding Shear Zone is predominantly characterized by deeply incised river valleys (Fig. 6a and profile 4 in Fig. 6b). To the northwest, the Runding Shear Zone terminates against the Cham Fault and borders the Cham Basin to the north (Fig. 6a). This contrasts to the southeast, where the Runding Shear Zone is expressed as a distinct slope break terminating against the Pfahl Shear Zone (Fig. 6a and profile 5 in Fig. 6c). Here, elevations of up to 1000 m a.s.l. occur to the northeast of the Runding Shear Zone, whereas generally lower elevations of less than 650 m a.s.l. prevail to the southwest. Interestingly, our

data also provide evidence for the presence of several lineaments splitting off the Runding Shear Zone towards the east and southeast. These lineaments rotate from an overall NW-SE into an E-W orientation, indicating the presence of a horsetail splay originating from the terminating Runding Shear Zone (Fig. 6a).

The Danube Shear Zone and Keilberg Fault Zone both show distinct slope breaks in topography (Fig. 6a). In the case of the Danube Shear Zone, however, this slope break is not expressed uniformly along-strike. Along the northwestern segment of the Danube Shear Zone, the slope break occurs very abruptly (Fig. 6a and profile 6 in Fig. 6c). In contrast, the central segment is largely buried beneath the Cenozoic sediments of the North Alpine Foreland Basin (profile 7 in Fig. 6c). Here, the topography is very irregular, with numerous embayments of Cenozoic sediments occurring across the Danube Shear Zone

and reaching far into the crystalline basement in the northeast (Fig. 6a). The boundary between the topographically well-expressed northwestern and the sediment-covered central segment of the Danube Shear Zone is defined by the Cham Fault (Fig. 6a). The southeasternmost segment of the Danube Shear Zone (also called "Aicha-Halser-Nebenpfahl") deviates from the course of the Danube River and separates a local basement high in the south from domain A1 in the north (profile 8 in Fig. 6c, c.f., Fig. 1).

The Cham Fault is also characterized by a well-defined topographic lineament (Fig. 7). In the south, the fault separates the elevated plateau and the sedimentary fill of the Bodenwöhr Trough in the west (domain A2) from the incised landscape of the Stallwang Fault Zone in the east (domain A1). The central part of the Cham Fault consists of at least two main segments cross-cutting the Pfahl Shear Zone and bordering the Cham Basin to the west (Fig. 7). At this location, the Runding Shear Zone terminates against the Cham Fault. In the north, the Cham Fault separates the Rötz Basin from an elevated and deeply

incised landscape in the eastern part of domain C2. Here, the fault is manifested as a single, distinct lineament, which can be traced at least up to the inferred eastern extension of the Luhe Line in the north (Fig. 7).

### 6.2.2  Statistical analysis of topographic lineaments

Detected lineaments mainly strike (I) NW-SE/WNW-ESE subparallel to the Pfahl, Danube, and Runding shear zones, and (II) NNW-SSE/N-S (Fig. 8a). Lineaments striking NNW-SSE/N-S are oriented subparallel to the Cham Fault and the Keil-

berg Fault Zone and are especially abundant in the northwestern part of domain A1, where subparallel river valleys are deeply incised into the mountainous landscape (Figs. 5b and 6). A third, subordinate ca. E-W striking direction observed in the lineament inventory is mainly related to the lineaments forming the proposed horsetail splay in the southeast of domain C1 (Figs. 6 and 8a).

Mapped faults predominantly strike NNW-SSE/N-S, which contrasts to the general NW-SE/WNW-ESE trend of the major

domain-bounding Pfahl, Danube, and Runding shear zones (Fig. 8b). This discrepancy can be related to the technical representation of major shear zones in the analyzed geological maps, being expressed as wide zones of fault rocks rather than by single lines (Teipel et al., 2008; Galadí-Enríquez et al., 2009b). Therefore, individual structural elements of the shear zones could not be included in the directional statistical analysis, which slightly biases the relative proportions of fault orientations.

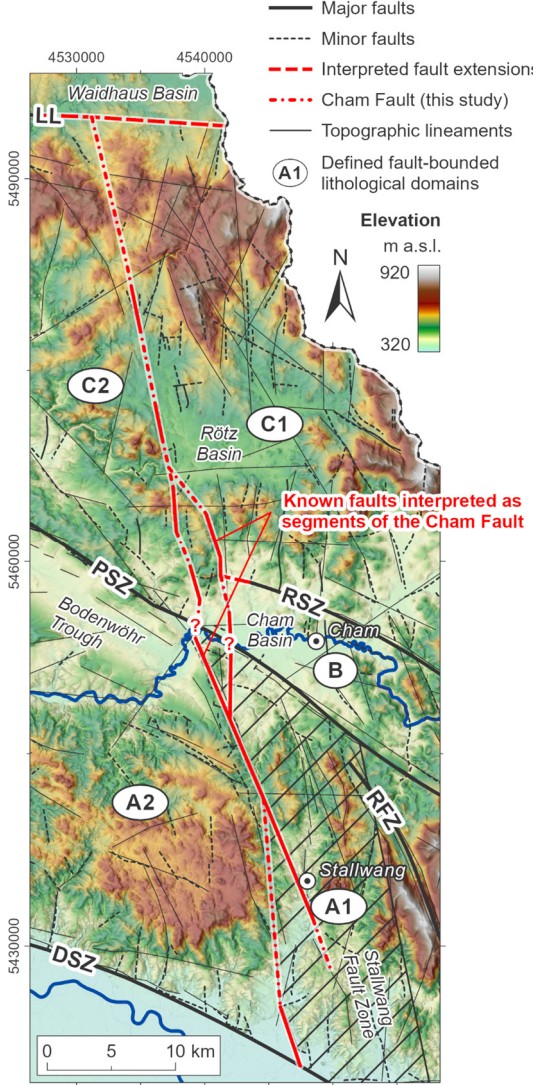

**Figure 7** Detailed map of the Cham Fault as inferred from the DEM. Several faults known from the geological map are interpreted as segments of the Cham Fault (faults from Teipel et al., 2008 and Galadí-Enríquez et al., 2009b). Note the uncertainty in the location of the southernmost part of the fault, which is due to enhanced erosion along the Stallwang Fault Zone. Fault zones: *DSZ* Danube Shear Zone, *LL* Luhe Line, *PSZ* Pfahl Shear Zone, *RSZ* Runding Shear Zone, *RFZ* Rattenberg Fault Zone.

Domains A1, A2, and C1 show distinct peaks in both NW-SE/WNW-ESE and NNW-SSE/N-S directions, similar to the overall trends of lineaments and faults in the study area (Fig. 8c). In contrast, domain B only shows a subordinate NW-SE to WNW-ESE oriented peak. Here, most of the lineaments are oriented NNW-SSE/N-S. Domain C2 shows a unique orientation pattern, with a distinct peak in the NW-SE/WNW-ESE directions, whereas N-S lineaments are almost entirely missing in this area (Fig. 8c).

The lineament densities also vary across the study area. Lineament densities to the west of the Cham Fault (domains A2 and C2, ca. 0.76 and 0.57 km/km², respectively) are lower in comparison to areas east of the Cham Fault (domains A1, B, and C1, between ca. 0.84 and 1.05 km/km²). Domain B shows the highest density of lineaments, reaching ca. 1.05 km/km² (Fig. 8c).

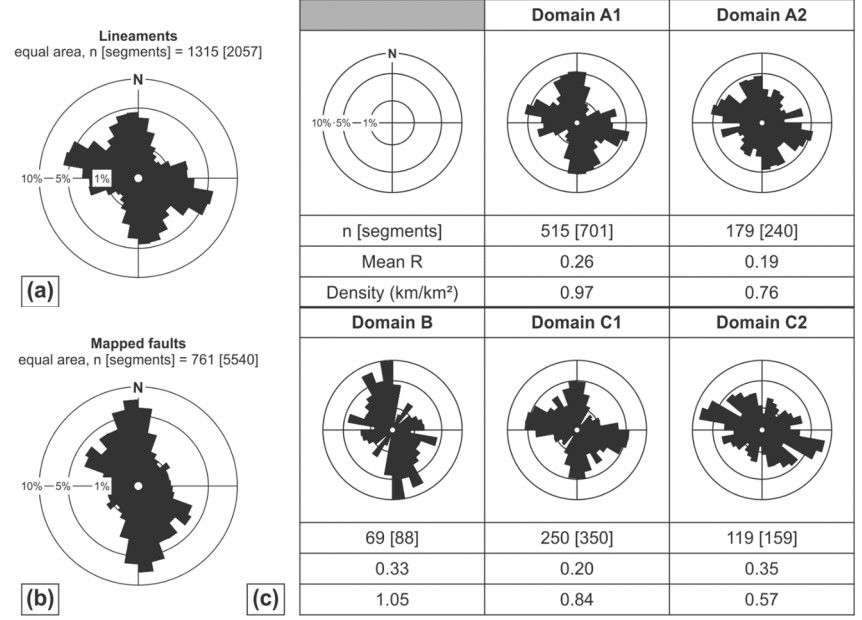

| | Domain A1 | Domain A2 |
|---|---|---|
| n [segments] | 515 [701] | 179 [240] |
| Mean R | 0.26 | 0.19 |
| Density (km/km²) | 0.97 | 0.76 |
| **Domain B** | **Domain C1** | **Domain C2** |
| 69 [88] | 250 [350] | 119 [159] |
| 0.33 | 0.20 | 0.35 |
| 1.05 | 0.84 | 0.57 |

**Figure 8** Rose diagrams of (**a**) lineaments detected in the DEM and (**b**) inferred faults from geological maps. (**c**) Statistical analysis of the detected lineaments carried out separately for each of the five defined domains. The analysis includes rose diagrams as well as calculations regarding the statistical scatter (mean resultant) and the mean spatial density of the lineaments. The mean resultant (*Mean R*) is a measure of dispersion analogous to the variance but expressed in the opposite sense. Lower values indicate a more dispersed distribution or the presence of different individual peaks, whereas higher values indicate that the observations are tightly bunched together (Davis, 2002). Only lineaments outside of a 2 km buffer along the domain boundaries were considered in the analysis to avoid biases related to the trends and lengths of the major fault zones and associated lineaments. Note that the Bodenwöhr Trough in domain A2 and the Cham Basin in domain B (c.f., Fig. 6) have been excluded from the analysis due to their sedimentary infill. All rose diagrams have been length-weighted and calculated using a bin size of 10°. Lineaments and faults have been segmented prior to statistical analysis.

## 7    Observations from published low-temperature thermochronological data

We compiled thermochronological data from the literature that were obtained on apatite and zircon and reflect the low-temperature history of the study area (Fig. 9). On either side of the Cham Fault, two Zircon Fission Track (ZFT) data points, located close to the village of Winklarn and ca. 2 km apart (Fig. 9b), record an apparent age gap of ca. 45 Myrs (ca. 260 Ma and 215 Ma, respectively, data from C.W. Naeser in Gebauer, 1984 and Wagner et al., 1997). Similar ages of ca. 250 Ma have been reported from a quarry to the south of the Luhe Line (northeastern part of domain C2) and a sample located in the central-eastern part of domain A2 (data from C.W. Naeser in Gebauer, 1984). In contrast, younger apparent ZFT ages of ca.

215 Ma are known from the Passau Forest, located close to the southern margin of domain A1 outside the study area, and from the area adjacent to the Teplá-Barrandian Unit (Domain C1, data from C.W. Naeser in Gebauer, 1984).

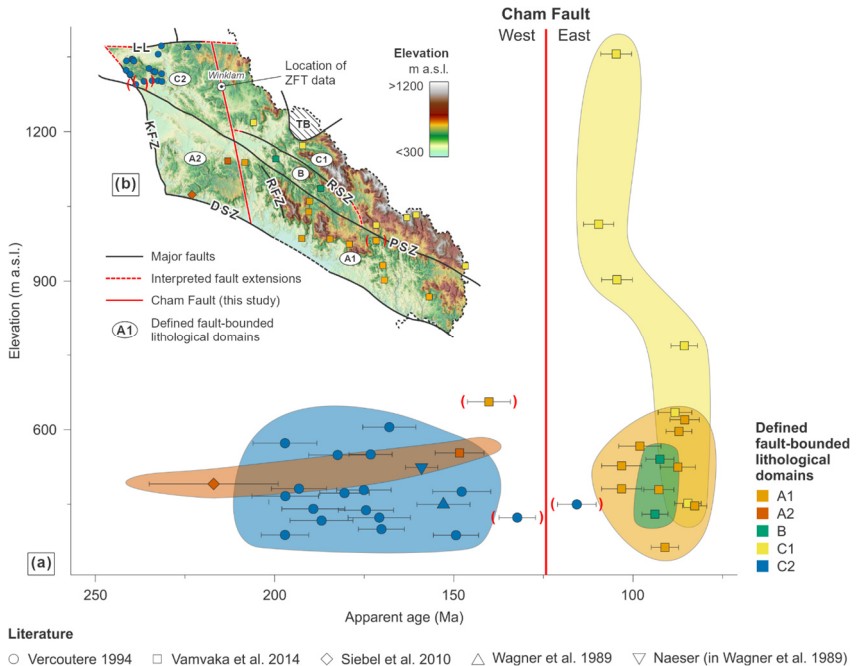

**Figure 9** Compilation of published Apatite Fission Track (AFT) data (Wagner et al., 1989; Vercoutere, 1994; Siebel et al., 2010; Vamvaka et al., 2014). (**a**) Age-elevation plot color-coded according to the sample's domain affiliation. Note that the Cham Fault separates two distinct clusters of AFT apparent ages. Calculation of the sample altitudes has been done based on a DEM with a 1-meter horizontal resolution using the sample location coordinates. (**b**) Locations of the samples shown in (**a**) superimposed on a DEM. Note that the village of Winklarn depicts the sample locations of published ZFT data, with one sample located at each side of the Cham Fault within a 2 km distance (see text for details). Fault zones: *DSZ* Danube Shear Zone, *KFZ* Keilberg Fault Zone, *LL* Luhe Line, *PSZ* Pfahl Shear Zone, *RSZ* Runding Shear Zone, *RFZ* Rattenberg Fault Zone. *TB* Teplá-Barrandian.

Low-temperature Apatite Fission Track (AFT) data also show significant spatial differences in apparent ages across the Cham Fault, even among samples obtained at similar altitudes (Wagner et al., 1989; Vercoutere, 1994; Siebel et al., 2010; Vamvaka et al., 2014; Fig. 9a). An apparent age gap of ca. 40 to 50 Myrs is recorded between two clusters separated by the newly defined Cham Fault. In the area to the west of the Cham Fault (domains A2 and C2), AFT apparent ages mainly cluster between ca. 150 and 200 Ma, whereas to the east of the fault (domains A1, B, and C1), most AFT data record apparent ages younger than ca. 100 Ma (Fig. 9a). No significant correlation between sampling elevation and ages is observed in the compiled AFT dataset. In addition, the data do not show a clear age gap across the Pfahl and Runding shear zones.

# 8 Discussion

By analyzing Bouguer gravity anomaly, topographic, and geological data, we identified three main basement domains in the southwestern Bohemian Massif. These domains are interpreted as individual basement blocks that were differentially exhumed during and after the late Paleozoic Variscan Orogeny. Furthermore, the heterogeneous distribution of exposed granites and available thermochronological data show a second segmentation by the previously unknown crustal-scale Cham Fault. Below we discuss the timing and succession of the observed block segmentation and its implications for the tectonic framework along the southwestern Bohemian Massif.

## 8.1 Lithological evidence for block segmentation

The observed metamorphic units in domain C are interpreted as being part of a genetic sequence of thermally overprinted rocks, starting from intermediate-grade biotite-garnet-chlorite schists close to the southern tip of the Teplá-Barrandian Unit over higher-grade mica gneisses to metatectic cordierite-sillimanite-K-feldspar gneisses in the adjacent area north of the Pfahl and Runding shear zones (Blümel, 1972; Fig. 3). Hence, a NNE to SSW progressive Buchan-type metamorphic zoning from upper greenschist facies to anatexis under low to intermediate pressures becomes apparent in the area north of the Pfahl and Runding shear zones (Read, 1952; Grauert et al., 1974; Winter, 2010). Consequently, the diatexites of domain A can be considered as successors in this sequence, where advanced partial melting of the metatectic progenitor has led to the formation of nebulitic and schlieren structures or, in parts, even to a completely homogenized, granite-like rock texture (Brown, 1973; Rohrmüller et al., 1996; Wimmenauer and Bryhni, 2007; Chen and Grapes, 2007). The metamorphic rocks of domain B are interpreted to have formed during an intermediate stage of anatexis compared to domains A and C.

Kyanite has not yet been observed in the anatectic rocks of the study area, thus bracketing pressure conditions to the andalusite-sillimanite stability field. This is in line with estimates of pressure and temperature conditions, which, unfortunately, are largely restricted to the area north of the Pfahl Shear Zone. Here, earlier studies estimated metamorphic conditions of 2-4 kbar and 650-730 °C (Schreyer et al., 1964; Schreyer and Blümel, 1974; Blümel and Schreyer, 1976, 1977). Higher P/T conditions of up to 5-7 kbar and 800-850 °C are suggested for the cordierite-bearing migmatites in the central part of the study area (Kalt et al., 1999).

A strong correlation between metamorphic grades and rock densities is observed, resulting in distinct variations of Bouguer anomalies between the identified domains. Higher-grade diatectic domains thereby show more pronounced negative Bouguer anomalies compared to lower-grade, gneissic domains, an observation that is in line with preliminary results of Schaarschmidt et al. (2019). Based on this relationship, domain B likely comprises rocks with a higher mean metamorphic grade ("diatectic gneisses") compared to the central part of domain A ("diatexites and gneisses"), important information that is not directly assessable from recent geological maps (Fig. 3). The distribution of diatectic gneisses in domain B and the presence of a distinct topographic lineament suggests a previously unknown extension of the southeastern tip of the Runding Shear Zone towards the south, terminating against the Pfahl Shear Zone (Figs. 3 and 6).

We assume differential vertical motions of the crust along distinct, basement block-bounding fault zones as the cause of the observed differences in exhumation and metamorphic grades (Fig. 10). The generally higher metamorphic grades in domain A indicate a higher total amount of exhumation southwest of the Pfahl Shear Zone, which is supported by variations in granite geochemistry across the fault (Grauert et al., 1974; Beer, 1981; Finger and Clemens, 1995; Siebel et al., 2008; Finger and Rene, 2009). In contrast, domain C, dominated by metatectic gneisses, is interpreted to have experienced a lower amount of exhumation. The intermediate grade metamorphic rocks in domain B indicate a lower amount of exhumation compared to domain A2 and the southeastern part of domain A1 but a higher amount of exhumation compared to domain C and the northwestern part of domain A1.

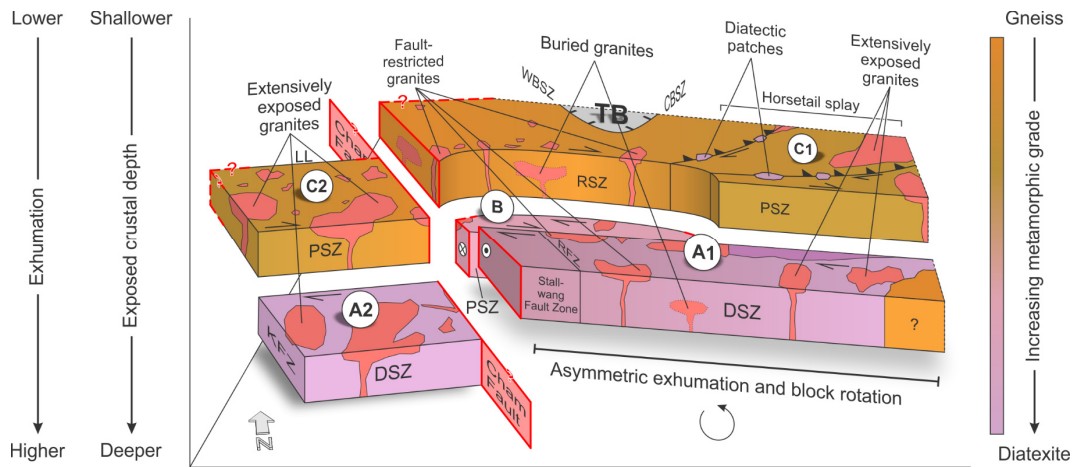

**Figure 10** Proposed three-dimensional block model of the southwestern Bohemian Massif. Distinct fault zones separate blocks of different lithological, gravity, and topographic characters, pointing to the exposure of different crustal levels and, thus, to varying amounts of exhumation, the latter increasing from domain C towards domain A. To the east of the Cham Fault, asymmetric exhumation has led to the tilting of the entire area towards the west. Note that the relative depth in the model shows the exposed crustal level, which is vice versa to exhumation and tectonic tilt. Fault zones: *CBSZ* Central Bohemia Shear Zone, *DSZ* Danube Shear Zone, *KFZ* Keilberg Fault Zone, *LL* Luhe Line, *PSZ* Pfahl Shear Zone, *RSZ* Runding Shear Zone, *RFZ* Rattenberg Fault Zone, *WBSZ* West Bohemia Shear Zone. *TB* Teplá-Barrandian.

A sharp contrast in the spatial distribution of exposed late Variscan granites from the northwest to the southeast is used as evidence for another level of segmentation along the Cham Fault. The Cham Fault separates areas exposing shallower crustal levels with less abundant granite exposures in the southeast (domains A1, B, and C1) from areas exposing deeper crustal levels and a higher number of granites in the northwest (domains A2 and C2, Fig. 3). From the Cham Fault to the southeast, granite exposure gradually increases. This observation is interpreted as a counterclockwise block rotation to the east of the Cham Fault (domains A1, B, and C1, Fig. 10). This rotation resulted in the exposure of deeper crustal levels in the very southeast of the study area, where both metamorphic grades and the amount of exposed late Variscan granites are similar to those observed in domains A2 and C2, respectively (e.g., diatexites in between the Fürstenstein and Hauzenberg composite massifs, Figs. 3 and 10).

Exposed late Variscan granites in the center of the study area are aligned with the Pfahl, Runding, and Danube shear zones. These tectonic structures appear to have guided magma ascent in this area, acting as low-pressure zones enabling magma transport to upper crustal levels as it is also observed, e.g., in the Alps and northeastern Brazil (Rosenberg, 2004; Weinberg et al., 2004). In the deeper subsurface, a nearly uniform distribution of granites is shown by the filtered Bouguer gravity anomaly data (Fig. 4), supporting early assumptions of Stettner (1975). Consequently, the (I) sporadic exposure of granites, (II) exclusive exposure of fault-related granites, and (III) presence of gneissic rocks within the central part of the diatexite-dominated domain A to the south of the Pfahl Shear Zone (in between the Cham Fault and the Fürstenstein Composite Massif, Fig. 3) can be explained by block rotation and differential exhumation.

## 8.2   Topographic evidence for block segmentation

The topographic analysis highlights a close relationship between metamorphic grades and topographic relief. Higher-grade metamorphic rocks (i.e., diatexites) thereby correlate with lower elevations, whereas lower-grade metamorphic rocks (i.e., gneisses) generally correlate with higher elevations (Fig. 5). This correlation is especially evident in the southeastern part of domain A1, where the occurrence of granites and diatexites is associated with a pronounced decrease in altitude (Fig. 5b). Hence, the observed correlation between topography and exposed rock type suggests that higher-grade metamorphic rocks are more prone to erosion compared to lower-grade metamorphic rocks of the study area. This relationship supports our interpretation of domain B comprising higher-grade metamorphic rocks compared to the central part of domain A (i.e., the northwestern part of domain A1), as it shows distinctly lower elevations compared to adjacent areas and thus catches the drainages of its surroundings. Nevertheless, it must be noted that also younger, late to post-Mesozoic differential tectonic block motion may represent an important factor controlling the recent distribution of elevated areas, as significant reactivation of Variscan structures has taken place during this time (e.g., Kley and Voigt, 2008; Voigt et al., 2021). In addition, spatial variations of joint and fracture densities are able to significantly affect the recent topography of the study area. Therefore, the observed correlation between higher-grade metamorphic rocks and lowered topography is most likely a local phenomenon that does not necessarily depict a general trend.

Lineaments detected based on topographic data along the southwestern Bohemian Massif greatly exceed both numbers and lengths of faults known from geological maps. This suggests the presence of a considerable number of yet unidentified faults or fault segments. From the statistical analysis, two main lineament directions (I) NNW-SSE/N-S and (II) NW-SE/WNW-ESE have been identified (Fig. 8a). These results are in accordance with previous studies of the lineament inventory along the southwestern Bohemian Massif (Lehrberger et al., 2003; Zeitlhöfler, 2007; Zeitlhöfler et al., 2015). The observed directions correspond to faults known from geological maps and are in line with the prevailing trend of the large, late Variscan fault zones, e.g., the Pfahl Shear Zone, Danube Shear Zone, and Keilberg Fault Zone (Fig. 8b). Therefore, a similar origin regarding timing and tectonic framework can be assumed for most of the detected structures. A third, ca. E-W oriented direc-

tion in the lineament inventory originates from the proposed horsetail splay located in the southeast of the study area. Such

an E-W orientation has only rarely been observed in mapped faults.

The tectonic configuration of the southwestern Bohemian Massif during the late stages of the Variscan Orogeny is interpret-
ed as a conjugate shear system related to N to NNW directed shortening (Wallbrecher et al., 1991; Brandmayr et al., 1995;
Peterek et al., 1996; Büttner, 1999; Galadí-Enríquez et al., 2010). Under this tectonic regime, a Riedel-type fault interaction

likely initiated the observed lineament patterns (Zeitlhöfler, 2007; Fig. 11). In such a model, NNW-SSE and WNW-ESE
striking lineaments would represent synthetic strike-slip faults with a right-lateral sense of motion (R- and P-Shears, respec-
tively). In contrast, ca. NNW-SSE to N-S striking lineaments either reflect antithetic strike-slip faults with a left-lateral sense
of shearing (R'-Shears, Fig. 11a) or extensional features that are oriented subparallel to the maximum principal stress (Fig.
11b). Our results confirm a Riedel-type fault interaction in domain B, which is tightly bracketed by the Pfahl and Runding

shear zones and shows by far the highest density of lineaments and fault rocks among the identified domains (Figs. 3 and
8c). The identified lineaments that split off the Runding Shear Zone in the southeastern part of domain C are interpreted as
fault branches forming a compressional horsetail splay (Figs. 10 and 11). By guiding the infiltration of fluids during meta-
morphism that lower the solidus and may trigger partial melting, as observed, e.g., in the Nanga Parbat Massif (Butler et al.,
1997), this horsetail splay and the associated shear zones may also be responsible for the occurrence of diatectic patches

arranged in a chain-like pattern in domain C (Figs. 3, 10, and 11).

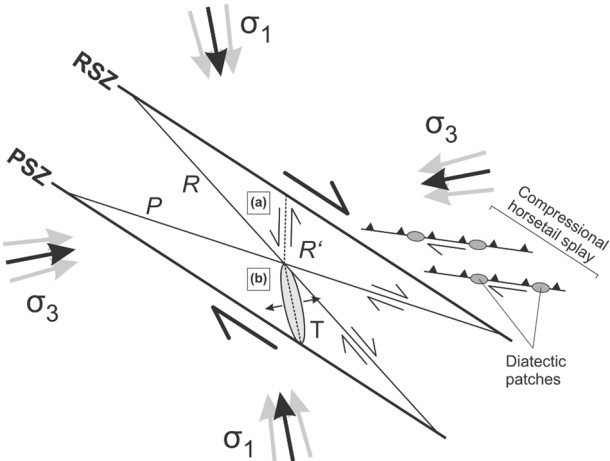

**Figure 11** Interpreted tectonic framework in domain B based on the orientations of detected topographic lineaments (modified and extend-
ed after Zeitlhöfler, 2007). NNW-SSE oriented "Riedel-Shears" (R) and WNW-ESE oriented P-Shears form as synthetic strike-slip faults
with a right-lateral sense of motion. Ca. NNW-SSE to N-S striking lineaments either represent (**a**) antithetic strike-slip faults with a left-
lateral sense of motion (R') or (**b**) extensional fractures (T), that are oriented subparallel to the maximum principal stress ($\sigma 1$). Fault zones:
*PSZ* Pfahl Shear Zone, *RSZ* Runding Shear Zone.

On a regional scale, the general structural configuration of the entire study area is also interpreted to reflect Riedel-type fault
interactions, which most likely initiated during late to early post-Variscan wrench tectonics (Peterek et al., 1996; Peterek et

al., 1997; Zeitlhöfler, 2007). During this time, an extensional basin is interpreted to have been situated along the western margin of the Bohemian Massif ("Naab Trough" sensu Schröder, 1988). Local outcrops of Carboniferous-Permian clastic sediments are thereby preserved along the northwestern segments of the Danube and Pfahl shear zones (e.g., in the Donaustauf and Schmidgaden basins). As these outcrops are entirely limited to the area west of the Cham Fault, we propose that the Cham Fault was active as a normal fault during late to early post-Variscan wrench tectonics that controlled the basin configuration during this time. This interpretation fits the reported ca. NNW-SSE to N-S oriented maximum principal stress during the late stages of the Variscan Orogeny (Wallbrecher et al., 1991; Brandmayr et al., 1995; Peterek et al., 1996; Büttner, 1999; Galadí-Enríquez et al., 2010) and is in accordance with Peterek et al. (1996), who speculated on the presence of ca. NNW-SSE to N-S oriented basin-bounding normal faults close to the eastern margin of the Bodenwöhr Trough. The presence of topographic lineaments crossing Variscan tectonic structures, however, indicates a younger (re)activation phase along the southwestern Bohemian Massif, which was most likely initiated in the course of Cretaceous to Cenozoic tectonics (Lehrberger et al., 2003).

### 8.3 Thermochronological evidence for block faulting

Low-temperature thermochronological ages obtained from Apatite and Zircon Fission Track analyses range from late Permian to Late Cretaceous and thus clearly postdate thermotectonic processes related to the late Paleozoic Variscan Orogeny. The spatial distribution of compiled data points shows that the newly identified Cham Fault separates two distinct clusters of FT ages.

From track length characteristics (mixed-bimodal to positively skewed), AFT data in the Naab Mountains to the west of the Cham Fault are interpreted to record post-Variscan cooling ages (ca. 270 Ma) that were partially reset during late Permian to Mesozoic subsidence and burial (up to 1000 to 1400 m) followed by Late Cretaceous to Cenozoic uplift and exhumation (Vercoutere, 1994).

An even more complex thermotectonic record is inferred for the Bavarian Forest in the central and southeastern part of the study area (Vamvaka et al., 2014). Here, thermal models indicate a first reheating during Late Jurassic times (ca. 160-140 Ma), which was followed by exhumation during the Early Cretaceous (ca. 140-120 Ma). After a phase of stagnation, sedimentation recurred during the Late Cretaceous (ca. 95-85 Ma), which caused reheating of marginal parts of the Bavarian Forest. The latter phase is especially depicted by the "Grub" sample, which is the only sample in the study of Vamvaka et al. (2014) that is located to the west of the newly identified Cham Fault (Fig. 9). Similar to the tectonic record of the Naab Mountains (Vercoutere, 1994), the final uplift phase of the Bavarian Forest was initiated in the Late Cretaceous (Vamvaka et al., 2014), probably in the course of inversion tectonics related either to the Alpine collision (e.g., Ziegler, 1987; Ziegler et al., 1995) or Africa-Iberia-Europe convergence (Kley and Voigt, 2008).

Hence, from this record, a complex regional thermotectonic evolution of the study area is proposed based on AFT data. Nevertheless, a significant difference in the thermal evolution between the sector to the west of the Cham Fault compared to the

eastern sector is undoubtedly present, as evidenced by two pronounced age clusters at either side of the fault (Jurassic vs. Late Cretaceous, Fig. 9). Vamvaka et al. (2014) attributed this change to the presence of a fault zone in between the Grub sample location and the remaining samples to the east (i.e., the Cham Fault), accommodating a vertical displacement of at least 1 km depending on the thickness of the Mesozoic sedimentary cover. This interpretation is in line with Gebauer (1984), who proposes the presence of a significant tectonic structure close to the village of Winklarn between domains C1 and C2, based on two ZFT data points located only 2 km apart. These record an age gap of ca. 45 Myrs (ca. 260 Ma and 215 Ma, respectively), which is again in accordance with the newly proposed Cham Fault.

A possible cause for the apparent differences in timing between the western sector (ZFT: late Permian, AFT: Jurassic) and the eastern sector (ZFT: Late Triassic, AFT: Late Cretaceous) might be attributed to the NW-SE prograding late Permian to Mesozoic depositional system (e.g., Meyer, 1989; Peterek et al., 1997; Schröder et al., 1997) in combination with differential exhumation across the Cham Fault. Based on the above-mentioned gap of ca. 45 Myrs between two ZFT data points close to the village of Winklarn, Meyer (1989) placed the basin margin during the Triassic close to this locality. From this interpretation, we conclude that also large parts of the southeastern margin of the Mesozoic basin most likely have been controlled by the Cham Fault, which, in turn, resulted in a larger sedimentary cover and the partial resetting particularly of AFT ages towards the west of the Cham Fault. This interpretation is in accordance with the AFT data towards the east of the Cham Fault, where most sample sites are thought to have been covered only with insignificant (<200 m) or, in parts, even completely lacking sediment thicknesses (Vamvaka et al., 2014). The younger AFT ages to the east of the Cham Fault could thereby be explained by enhanced exhumation during Late Cretaceous inversion tectonics (e.g., Kley and Voigt, 2008; Voigt et al., 2021). As evidenced by the thermochronological record, thrusting of basement blocks led to km-scale exhumation of numerous basement domains in Central Europe during this time (von Eynatten et al., 2021). The generally higher elevation to the east of the Cham Fault might be a remnant of this tectonic impulse. In addition, the NW-SE increasing thicknesses of Upper Cretaceous sediments in the "Regensburg-Straubing Trough" in front of the Danube Shear Zone (Führer, 1978; Unger and Risch, 1991) imply a more intense reactivation of the southeastern segment of the Danube Shear Zone. This also supports an enhanced Late Cretaceous exhumation of the adjacent area to the north of this fault segment (i.e., domains A1, B, and C1). If we assume only an insignificant sedimentary cover during the Late Cretaceous in the study area (≪1 km), subsequent vertical displacement along the Cham Fault must have been greater than 1 km (Vamvaka et al., 2014) and was probably in the range of 1.5 to 2 km.

In contrast to the Cham Fault, the Pfahl and Runding shear zones do not delimit areas of different thermochronological histories. Here, only minor vertical tectonic movements across present faults are made responsible in post-Cretaceous times for varying altitudes among samples showing similar AFT ages (Vamvaka et al., 2014).

### 8.4 Relative timing and succession of block segmentation events

ZFT data indicate unroofing of the western Bohemian Massif during Permian and Triassic times (Hejl et al., 1997). This is supported by Permian syn-tectonic sedimentation in the Donaustauf Basin bounded by the northwestern segment of the Danube Shear Zone (Siebel et al., 2010). K-Ar and Rb-Sr illite ages thereby bracket uplift and erosion of the Variscan basement and tectonic extension to a period prior to 255-266 Ma (Siebel et al., 2010).

From the lithological and thermochronological data, at least two phases of post-Variscan fault movement and basement block segmentation along the southwestern Bohemian Massif can be inferred: (I) Vertical displacements along the major NW-SE striking Pfahl, Danube, and Runding shear zones, segmenting the area into NW-SE trending blocks of different metamorphic grades ("Pfahl Phase"), and (II) tectonic movements along the Cham Fault, being responsible for further block segmentation and counterclockwise block rotation ("Cham Phase", Fig. 10). Contrasting cross-cutting relationships between the major faults and the late Variscan granites, however, provide a challenge to reconstruct the exact timing of these two phases:

(I) Pfahl Phase: The Pfahl and the Runding shear zones cross-cut (Patersdorf Stock ca. 323 Ma: Siebel et al., 2006a and Arnbruck Stock ca. 325 Ma: Siebel et al., 2008) or, in some places, truncate granites (Rinchnach Stock ca. 320-329 Ma: Siebel et al., 2006a and Haidel Massif ca. 323 Ma: Siebel et al., 2008; Table 1; Fig. 3). Together with a distinct mylonite zone within the Patersdorf Stock, this indicates an activity along the Pfahl and Runding shear zones post-dating the emplacement of late Variscan granites (Siebel et al., 2006a; Büttner, 2007). This is further supported by quartz mineralizations within the mylonite zone ("Pfahl Quartz") of the Pfahl Shear Zone, indicating a Triassic re-activation of the shear zone (Horn et al., 1986). In contrast, numerous granites appear to have utilized the NW-SE striking fault zones for their ascent, indicating that the faults already existed during granite emplacement. Additionally, there is indication that post-Variscan faulting and shear zone formation in the exhumed basement follows pre-existing structures formed in the deeper continental crust during a late stage of the Variscan Orogeny prior to granite emplacement. Linear intrusions of I-type granodiorites containing abundant mafic enclaves ("Palit-Komplex", Fig. 3) at 334 Ma (Siebel et al., 2005) delineate weak zones in the deeper crust that have been overprinted by the Pfahl shear zone during and after the basement exhumation.

(II) Cham Phase: The exposure of late orogenic granites limited to the west (most likely the hanging-wall) of the Cham Fault suggests post-granitic faulting as the fault sharply cuts the granite inventory. Erosional products of late orogenic granites deposited in adjacent Carboniferous-Permian basins indicate rapid exhumation during the Cham Phase shortly after late Carboniferous granite emplacement (Welzel, 1991; Mielke, 1993; Galadí-Enríquez et al., 2009a).

As it becomes apparent from the time constraints of fault activity and block segmentation mentioned above, the Pfahl Phase cannot be considered as a single event but rather as a series of events, which, in total, resulted in a higher total amount of exhumation towards the southwest of the Pfahl Shear Zone (domain A). Total exhumation during the Pfahl Phase must have amounted to less than the mean 14-18 km granite emplacement depths (Table 1). Otherwise, a homogeneous distribution of granites, as observed in domains A2 and C2 and in the deeper subsurface, would be expected across the surface of the entire study area (Fig. 4). In contrast, the Cham Phase is considered as post-granitic and involved higher amounts of exhumation in the northwest (domains A2 and C2) and, likely due to block rotation, in the southeast of the study area. This segmentation phase resulted in the area-wide exposure of granite bodies at the Earth's surface, with total exhumation during both segmentation phases (i.e., the Pfahl and Cham phases) exceeding the mean granite emplacement depths of 14-18 km (Table 1).

## 9    Conclusions

Our integrated study using gravity anomaly and topographic data provides insights into the crustal architecture of the southwestern Bohemian Massif. Domains of different metamorphic grades or exposed granite inventories are interpreted as individual crustal blocks that are bordered by distinct tectonic structures. With the Cham Fault, we introduce a previously unknown NNW-SSE striking tectonic structure that is of similar importance to other major fault zones, such as the Pfahl and Danube shear zones. We propose a model of differential exhumation and block rotation that led to the juxtaposition of contrasting lithological domains. Increasing amounts of exhumation evoked the exposure of deeper crustal levels, as evidenced by the presence of higher metamorphic grades and a higher percentage of granite bodies at the surface. Gravity anomaly filterings thereby indicate a rather homogeneous distribution of granites in the subsurface. This observation contrasts with the heterogeneous exposure of granites at the surface, suggesting that an important phase of segmentation and differential exhumation must have occurred after granite emplacement. The post-Variscan activity of the Cham Fault is evidenced by abrupt changes of apparent ZFT and AFT ages across the tectonic structure. The fact that the Cham Fault also forms tectonic boundaries of Cretaceous to Cenozoic geological features, such as the Bodenwöhr Trough, implies that it played a significant role in the tectonic evolution of the southwestern Bohemian Massif even during the younger geological past.

Our model of block segmentation and differential exhumation along the southwestern Bohemian Massif emphasizes the significance of vertical displacements along distinct tectonic structures to explain the observed complex lithological configuration in the study area. The example of the newly discovered Cham Fault thereby suggests that potentially several more yet unidentified tectonic structures might exist in that area. To precisely reconstruct the timing and succession of block segmentation events along the discussed faults and to quantify the amounts of exhumation of the five outlined basement blocks, however, additional data on granite intrusion depths, P-T metamorphic conditions, thermochronology, and fault kinematics are needed.

**Data availability.** The high-resolution topographic data used in this study are available by request at the Bavarian Agency for Digitisation, High-Speed Internet and Surveying. The Bouguer anomaly data are provided by the Leibniz Institute for Applied Geophysics (LIAG).

**Author contribution.** AE, HF, WB, and HS designed the study. AE prepared the manuscript and performed the analysis of the topographic, filtered gravity, granite density, and literature data. HF carried out the filtering of the gravity data and supervised the study. HS and HdW supervised and acquired the financial support for the project leading to this publication. GG provided the gravity data and supported their interpretation. All authors contributed to the reviewing of the manuscript and

the discussion of the results.

**Competing interests.** The authors declare that they have no conflict of interest.

**Acknowledgements.** This research was part of the project "Lithologische und strukturelle Untersuchungen im ostbayerischen Grundgebirge", which was funded within the scope of the project initiative "Bodenatlas Bayern" by the Government of Bavaria with co-funding of the European Union (EFRE-Programm Bayern 2014-2020). We thank Caroline Vercoutere and Agni Vamvaka for providing details on their AFT analyses and Johann Rohrmüller, Timo Spörlein, Volker Friedlein,

Melanie Meyer (all LfU Bayern), Anna Schaarschmidt, and Tobias Stephan for fruitful discussions. Wolfgang Siebel and an anonymous reviewer are thanked for their valuable comments and suggestions during the review process.

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
