# Peer review of "Late to post-Variscan basement segmentation and differential exhumation along the SW Bohemian Massif, Central Europe"

_Solid Earth, 2021_

## Referee Comment (RC1)

[referee-annotated manuscript omitted]

---

## Author Comment (AC1)

**Response letter to RC1**

We would like to greatly appreciate the comments and suggestions made by referee 1, Dr. Wolfgang Siebel, that increased the scientific quality of the manuscript. Below are our replies to Dr. Siebel comments.

**Comments made in the manuscript pdf**

**Comment 1**, line 63
"Justification"

> Response from authors:
> *We have reorganized this sentence to include the most important pieces of evidence for widespread faulting and HT/LP metamorphism during the late stages of the Variscan Orogeny (e.g., the presence of large igneous domains and the complex juxtaposition of high- and low-grade metamorphic domains).*

**Comment 2**, line 79
"igneous rocks"

> Response from authors:
> *Changed accordingly.*

**Comment 3**, line 85
"it should be mentioned that this area also contains remnants of a recently identified island arc (Propach et al. 2008)"

> Response from authors:
> *We added a description regarding these findings.*

**Comment 4**, line 127
"phrasing
 better: structural pattern"

> Response from authors:
> *Changed accordingly.*

**Comment 6**, line 210
"note, the southeastern part of this domain almost exclusively consist of igneous rocks: Hauzenberg, Fürstenstein and orthoanatexite complex in between. So, this part differs completely from the remainder of A1"

> Response from authors:
> *As there are several comments on this topic (subdivision of domain A), we provide a detailed response at the end of this response letter (below comment 2 of "Specific remarks not made in the manuscript pdf", which heads in the same direction).*

**Comment 7**, lines 223-224
"as mentioned above this part almost entirely consist of igneous and metamorphosed igneous rocks (would make sense to separate this part from domain A)"

> Response from authors:
> *Please see our response to comment 6. We provide a detailed response at the end of this response letter (below comment 2 of "Specific remarks not made in the manuscript pdf", which heads in the same direction).*

**Comment 8**, Figure 3

"between FST and HZB is a large orthoanatexite terrane (see Propach et al. 2008), Siebel et al. 2012) reminiscent to the palites and without relation to the metasedimentary rocks (diatexites, gneisses) of unit A1"

> Response from authors:
> *Please see our response to comment 6. We provide a detailed response at the end of this response letter (below comment 2 of "Specific remarks not made in the manuscript pdf", which heads in the same direction).*

**Comment 9**, line 245

"cordierite plays an important role in these rocks. Part ot the metatextic rocks contain more than 20 % cordierite! Please add this point"

> Response from authors:
> *We added this point.*

**Comment 10**, line 247

"C into C1 and C2 ? (because here you are discussing domain C"

> Response from authors:
> *Here, we intend to emphasize that the NNW extension of the boundary subdividing A into A1/A2 also forms the boundary line between C1 and C2. We clarified this explanation in the revised text.*

**Comment 11**, line 256

"this is seen also in fig. 15 of Schaarschmidt et al. (2019). Please add this reference"

> Response from authors:
> *We added this reference.*

**Comment 12**, lines 291-292

"this observation supports the special status of this igneous subdomain and would justify its separation from domain A sensu stricto (see comments above)"

> Response from authors:
> *Please see our response to comment 6. We provide a detailed response at the end of this response letter (below comment 2 of "Specific remarks not made in the manuscript pdf", which heads in the same direction).*

**Comment 13**, lines 326-327

"by now, ridge remnants are only left at two sides that escaped the quartz mining:  quartz reefs at Viechtach and Regen

> Response from authors:
> *We agree that distinct, steep quartz reefs are only left at Viechtach and Regen. Nevertheless, many other segments of the Pfahl Shear Zone are still visible as, although subtle, ridges in topography, which might be due to the remnant quartz mineralization in the subsurface.*

**Comment 14,** line 383

"you should say that the lineaments were recognized based on topographic feature analyses and not on field mapping (see also line 479)"

> Response from authors:
> *We clarified this.*

**Comment 15,** Figure 9

"why not shown full rose diagrams for the different domains such as in figure 8. I guess this would be more legible"

> Response from authors:
> *We now show full rose diagrams in the revised manuscript.*

**Comment 16,** Figure 9

"looks that this subfigure presents the same data as figure 8a - so this plot is redundant"

> Response from authors:
> *We have merged figures 8 and 9 to avoid redundancy.*

**Comment 17,** line 389

"detected"

> Response from authors:
> *Changed accordingly.*

**Comment 17,** Figure 10

"Naeser (in Gebauer, 1984)?"

> Response from authors:
> *The ages were first precisely mentioned in Wagner et al. (1989), which is why we give this reference. In contrast, ZFT data, which have also been measured by Naeser, have already been mentioned in Gebauer (1984).*

**Comment 18,** line 433

"odd phrasing - please replace this term by e.g., progenitor / source material or else"

> Response from authors:
> *Changed in "progenitor".*

**Comment 19,** lines 470-471

"unclear sentences"

> Response from authors:
> *With this sentence, we intend to highlight the special lithological character of the area in between the Patersdorf Stock and the Metten Massif (or in between the Cham Fault and the Fürstenstein Composite Massif, respectively). Unlike the northwestern and southeastern parts of domain A, this central part is not only formed by diatexites but also contains intercalations of gneissic rocks, which we interpret as remnants that escaped anatectic overprint. Together with the fact that granites in this part of domain A occur exclusively along the major shear zones (e.g., Patersdorf Stock and Metten Massif), this points to a lower amount of exhumation and a westward directed tilt of the domains A1, B, and C1. We clarified this explanation.*

**Comment 20,** lines 474-475

"why should this be the case?"

> Response from authors:
> *We agree that this conclusion might appear a little overinterpreted. Nevertheless, the spatial correlation between higher-grade metamorphic rocks (i.e., diatexites) correlating with lowered topography and lower-grade metamorphic rocks (i.e., gneisses) generally correlating with higher elevations is evident (c.f., Fig. 5 b and c). We therefore adjusted this sentence so that it only gives indications for the above-mentioned relationship.*

**Comment 21,** Figure 12

"meaning of "P" and "R" needs to be explained"

> Response from authors:
> *We added this explanation to the figure caption.*

**Comment 22,** line 515

"be more precise, what ages do you mean? ZFT or AFT or both?"

> Response from authors:
> *Here, we mean both types of FT ages. We clarified this part.*

**Comment 23,** line 548

"unclear statement: restricted exposure could also mean in limited quantity. I guess you want to say than granites mainly occur west of the Chan fault, e.g., in the Regensburger Wald and in the southern Oberpfälzer Wald. Please make it more clear."

Response from authors:

*Indeed, here we want to say that granites mainly occur west of the Cham Fault. We clarified this part.*

In addition to the above listed comments, we corrected all technical errors mentioned in the manuscript pdf (lines 331, 495, 504, 510, 512, 548, 559 …)

**Specific remarks not made in the manuscript pdf**

**Comment 1**

"A Bouguer anomaly map was already published and discussed by Schaarschmidt et al. (2019 Figure 15). Their figure shows clear differences in gravity anomaly between the Vorderer and Hinterer Bayerischer Wald and areas of granite accumulations and such previous results should be adequately mentioned in the present manuscript."

Response from authors:

*We now refer to previous results of Schaarschmidt et al. (2019) when describing the different gravity signatures of the "Vorderer Bayerischer Wald" (domain A) and the "Hinterer Bayerischer Wald" (domain B and C). Schaarschmidt et al. (2019) presented preliminary results based on a qualitative discussion of the observed, unfiltered Bouguer anomalies; we applied a high-pass filtering technique and have a stronger focus on the detection of buried granite bodies. The gravity map shown in Schaarschmidt et al. (2019) depicts gravity anomalies of the entire crust and, therefore, it reflects superimposed gravity signatures of different crustal units at various depths. The short-wavelength part of the filtered anomaly, as used in our study, generally better correlates with (near-)surface geology. Hence, an unfiltered gravity map does not differentiate between the signature of rock units located at different crustal depths nor between lower density rocks, e.g., granites and sedimentary rocks*

**Comment 2**

"At line 85 and for reconsidering the subdivision of domain A (see comments in pdf file), the following reference should be considered more strongly: Propach G, Kling M, Lindhardt E, Rohrmüller J (2008) Remnants of an island arc within the Moldanubian zone of the Bavarian Forest. Geologica Bavarica 110: 343–377"

Response from authors:

*Thank you very much for your suggestion to reconsider the subdivision of domain A. In this paper, we aim to relate the lithological expression of the southwestern Bohemian Massif to its structural architecture, therefore we only have included bounding structures that are presumably of tectonic origin (e.g., curvilinear to linear boundaries, such as the Pfahl Shear Zone and the inferred Cham Fault). In the case of the southeastern part of domain A, we could not define such a distinct boundary. Identified domains are fault-bounded lithological units, while in the southeastern part of domain A no fault could be interpreted based on geological maps and presented lineament analysis, hence we interpret the southeastern part of domain A as a lithologically different portion of the same domain.*

*The mentioned reference of Propach et al. (2008) indeed highlights the special status of the southeastern part of domain A. However, from the current metamorphic expression of domain A, it rather appears as a gradual transition from diatexites with intercalated gneissic rocks (see reply to comment 19) and fault-restricted granites in the central part of domain A to homogenous ortho-diatexites and extensively exposed granites in the southeast.*

*The transition between high topography in the central part of domain A and lowered topography in the southeast appears to be rather gradual and does not correlate with a single distinct topographic lineament. Instead, the transition from high to low elevations appears to be closely linked to the occurrence of granites and ortho-anatexites in between the Fürstenstein and Hauzenberg composite massifs. This suggesets that the topographic expression in this part of the study area is rather a result of varying rock erodabilities that appear to depend on the metamorphic grade of the exposed rock units (c.f., comment 20). From the filtered Bouguer anomaly map (Fig. 4 c and d), domain A1 is defined by the same*

*pattern of alternating high and low gravity signals throughout its entire extent, pointing towards a similar metamorphic configuration in the subsurface. This contrasts to domain A2, which is characterized by a much lower amplitude in the filtered gravity signal.*

*Consistent Fission-Track ages in domain A further support our view that a km-scale tectonic structure separating the southeastern part of domain A from the rest is most likely missing.*

*Nevertheless, we agree that this interpretation might be less comprehensible compared to the interpretation of the Cham Fault. Therefore, we added a paragraph explaining why we decided not to include another domain in this part of the study area. We have also replaced the term "lithological domain" with "fault-bounded lithological domain" to clarify its definition.*

**References**

Gebauer, D.: Erdgeschichtliche Entwicklung und geologischer Überblick, in: Erl. Geol. Kt. Bayern 1:25.000, Bl. 7446 Passau, edited by: Bauberger, W. and Unger, H. J., München, 13–22, 1984.

Propach, G., Kling, M., Linhardt, E., and Rohrmüller, J.: Remnants of an island arc within the Moldanubian zone of the Bavarian Forest, in: Geologica Bavarica Nr. 110: Geochronologische, geochemische, petrographische und mineralogische Untersuchungen im Grundgebirge Bayerns sowie kritische Betrachtungen zu Sr-Isotopenstandards, edited by: Bayerisches Landesamt für Umwelt, Augsburg, 343–377, 2008.

Schaarschmidt, A., Haase, K. M., de Wall, H., Bestmann, M., Krumm, S., and Regelous, M.: Upper crustal fluids in a large fault system: microstructural, trace element and oxygen isotope study on multi-phase vein quartz at the Bavarian Pfahl, SE Germany, Int. J. Earth Sci. (Geol. Rundsch.), 108, 521–543, https://doi.org/10.1007/s00531-018-1666-y, 2019.

Wagner, G. A., Michalski, I., and Zaun, P.: Apatite Fission Track Dating of the Central European Basement. Postvariscan Thermo-Tectonic Evolution, in: The German continental deep drilling program (KTB): Site selection studies in the Oberpfalz and Schwarzwald, edited by: Emmermann, R., Springer, Berlin, Heidelberg, New York, London, Paris, Tokyo, Hong Kong, 481–500, 1989.

---

## Author Comment (AC2)

**Response letter to RC2**

We appreciate the critical and constructive comments and suggestions made by anonymous referee 2, that increased the scientific quality of the manuscript. Below are our replies to anonymous referee 2 comments.

**Comment 1**

"Some terminology is not properly used and can be misleading. I know that it is a boring and common debate, but terms uplift, exhumation and erosion should be used properly. For instance metamorphic data are usually used to quantify exhumation, i.e. the vertical movement with respect to the earth surface. It sounds strange quantify uplift by metamorphic condition. Same approach should be used with thermochronological data."

Response from authors:
*Thank you for pointing out the imprecise use of the terms "uplift, exhumation, and erosion". We have checked all of these terms for proper usage and revised the manuscript accordingly.*

**Comment 2**

"One of the my main criticism is in using the metamorphic degree as a tools to quantify differential exhumation of crustal blocks. The authors should review the different mineral assemblages that characterize each domain and evaluate the pressure condition that in this case can be eventually used to evaluate the depth of metamorphic event. I think that metamorphic degree only is not enough to discriminate the depth of the metamorphic event, especially in this case where differences of exhumation are proposed between domain of high metamorphic degree, e.g. between diatectic gneisses and diatextites."

Response from authors:
*Thank you very much for giving us the chance to clarify why we think using the metamorphic grade to reveal first-order differences in the relative amount of exhumation is appropriate. Below, we provide a brief summary of the metamorphic configuration of the study area. Subsequently, we discuss the different models that could explain this metamorphic configuration and why we think that a tectonically-driven model is the most plausible.*

*From the geological map, the study area can be divided into three first-order domains, each of which is characterized by a distinct pattern of metamorphic overprint (domains A, B, and C, Fig. 3 in manuscript). Identified domains are sharply separated by fault zones, such as the Pfahl and Runding shear zones. A general NNE to SSW increase in the metamorphic grade is observed in the study area. In the north (domain C), mica schists and mica gneisses occur close to the southeastern border of the Teplá-Barrandian unit (Fig. 3 in manuscript). Farther south, the onset of anatexis is indicated by metatectic cordierite-sillimanite-K-feldspar gneisses. An abrupt increase in the anatectic grade occurs along the Runding Shear Zone, as indicated by the presence of diatectic, garnet-bearing gneisses in between the Runding and Pfahl shear zones (domain B, Fig. 3 in manuscript). Another jump in the anatectic grade occurs along the Pfahl Shear Zone, as evidenced by vast complexes of diatexites in between the Pfahl and Danube shear zones (domain A, Fig. 3 in manuscript).*

*To explain the metamorphic zoning presented above, we postulate a tectonic model involving differential exhumation of distinct crustal blocks (domains A, B, and C), with progressively deeper crustal levels exposed towards the southwest. In fact, such a model of deeper crustal levels exposed to the southwest of the Pfahl Shear Zone is not novel and has been frequently assumed in previous work (e.g., Grauert et al., 1974; Beer, 1981; Finger and Clemens, 1995; Bader, 1996; Siebel et al., 2008; Finger and Rene, 2009) but the possible reason for such abrupt changes in metamorphic grades observed in the study area is presented for the first time in this study.*

*We agree that a detailed analysis of the mineral assemblages within the three outlined domains is required to evaluate the pressure conditions under which metamorphism occurred and ultimately to quantify the amount of differential exhumation. Table 1 summarizes the different mineral assemblages of the metamorphic rocks, including index minerals and textures. For details on the mineral assemblages and the inferred metamorphic grades, the reader is referred to the original publications referenced in Table 1.*

Table 1 Sequence of predominant metamorphic rocks in the study area from NNE to SSW. Index minerals and textures are indicated (data from Blümel and Schreyer, 1977; Baburek, 1995; Bader, 1996; Mielke, 1996; Rohrmüller et al., 1996; Kalt et al., 1999; Teipel et al., 2008; Propach et al., 2008). Mineral abbreviations: *And* andalusite, *Bt* biotite, *Chl* chlorite, *Crd* cordierite, *Grt* garnet, *Hbl* hornblende, *Kfs* K-feldspar, *Opx* orthopyroxene, *Sil* sillimanite.

| Predominant rock type | Index minerals | Textures | Domain | |
|---|---|---|---|---|
| Mica schist | And, Bt, Chl, Grt | Foliated | | NNE |
| Mica gneiss | And, Bt, Crd, Sil | Foliated | C | |
| Metatectic gneiss | Bt, Crd, Grt, Kfs, Sil | Leucosome schlieren | | |
| Diatectic gneiss | Bt, Crd, Grt, Kfs, Opx, Sil | Metablastesis | B | *Runding S.Z.* |
| Diatexites | Bt, Hbl, Kfs, Opx, Sil | Nebulitic / schlieren to homogeneous | A | *Pfahl S.Z.* SSW |

*From the arrangement of metamorphic rocks and their different mineral assemblages presented in Table 1, a NNE to SSW progressive Buchan-type metamorphic zoning from upper greenschist facies to anatexis under low to intermediate pressures becomes apparent (Read, 1952; Grauert et al., 1974; Winter, 2010). Kyanite has not yet been observed in the anatectic rocks of the study area, thus bracketing pressure conditions to the andalusite-sillimanite stability field. This is in line with estimates of pressure and temperature conditions, which, unfortunately, are largely restricted to the area north of the Pfahl Shear Zone. Here, early studies estimated metamorphic conditions of 2-4 kbar and 650-730 °C (Schreyer et al., 1964; Schreyer and Blümel, 1974; Blümel and Schreyer, 1976, 1977). Higher P/T conditions of up to 5-7 kbar and 800-850 °C are suggested for the cordierite-bearing migmatites in the central part of the study area (Kalt et al., 1999).*

*To precisely evaluate the different pressure conditions in each of the domains, however, a much greater dataset compared to the one presented in Table 1 is needed. Unfortunately, such a dataset does not yet exist for the study area. In fact, even with the availability of such a dataset, several problems would arise from its interpretation. First, due to the highly anatectic character of the entire study area, pressure differences among the identified domains are most likely in a very narrow range, probably even within the range of the statistical/methodological error. High-temperature metamorphic overprint is thought to has occurred in very shallow depths (<20 km, Kalt et al., 1999), which further complicates quantifying contrasts in pressure conditions. In addition, the problem arises that in the range of the inferred peak metamorphic conditions (>700 °C) further changes in the pressure-indicative mineralogy of the rocks are nearly absent, as significant partial melting is already taking place (Winter, 2010). Instead, progressive anatexis, e.g., due to the increased temperatures in the deeper crust, rather results in textural changes, such as the formation of schlieren and nebulitic structures, which are widely observed in the study area (Table 1; e.g., Brown, 1973; Wimmenauer and Bryhni, 2007; Chen and Grapes, 2007; Rohrmüller et al., 1996). From the presence of anatectic rock textures (e.g., schlieren and nebulitic structures) and the occurrence of orthopyroxenes, the highest anatectic conditions certainly occurred towards the south of the Pfahl Shear Zone. Four endmember scenarios could thereby explain the described metamorphic zoning in the study area:*

*(I)   Lateral differences in protolith rocks with varying solidus*
*(II)  Lateral differences in temperature during metamorphism*
*(III) Lateral differences in the amount of available fluids, e.g., due to different contents of OH-bearing minerals or the presence of fluid-pathways*
*(IV) Post-metamorphic, tectonically-driven differential exhumation of crustal blocks leading to contrasting anatectic domains at the present level of erosion (this study)*

*Below, we discuss the four different scenarios that could potentially explain the encountered metamorphic configuration in the study area and why we think that a tectonically-driven model (IV) is the most plausible.*

*The protoliths of the anatectic rocks in the Bavarian part of the southwestern Bohemian Massif are considered as a monotonous sequence of mainly pelitic greywackes ("Monotoneous Group", e.g., Rohrmüller et al., 1996). Only in the Passau Forest (southeastern part of domain A), a significant area is formed by amphibole-bearing diatexites, pointing to a greater abundance of igneous protoliths and to the existence of a former island arc (Propach et al., 2008). A sharp contrast in the protoliths of the three*

*outlined domains, similar to the tectonic boundary between the Austrian Ostrong unit (= Monotoneous Group) and Drosendorf unit (= Varied Group) (Fuchs, 1995), however, has not yet been identified in Bavaria (Propach et al., 2008). Therefore, a scenario of different protolith rocks in each of the three domains (Scenario I) most likely is not plausible.*

*Regarding the mechanisms of heat supply during the Variscan metamorphic event, models range from magmatic underplating (Kalt et al., 1999; Kalt et al., 2000) to lithospheric delamination (Klein et al., 2008). Regardless of the true mechanism, however, a sharp lateral temperature boundary (Scenario II), which would be required to explain the observed anatectic segmentation, is very unlikely.*

*A conceivable mechanism that could trigger anatexis in spatially limited areas is the presence of fluids (Scenario III). Fluids could thereby be provided either internally by the breakdown of OH-bearing minerals (e.g., micas; Le Breton and Thompson, 1988) or externally via fluid-pathways ("fluid-enhanced", e.g., Prince et al., 2001). As discussed previously, no discernible differences in the protoliths among the three defined domains have been discovered yet. Therefore, a model of different amounts of OH-bearing minerals in the protoliths of the domains is less plausible. In contrast, varying amounts of externally derived fluids, for example, infiltrated through tectonic structures such as shear zones, appear to be a reasonable mechanism to explain sharp contrasts in the anatectic grade. Indications for such an influx of $H_2O$ along tectonic structures triggering enhanced partial melting are provided by the presence of diatectic patches aligned with tectonic structures of the horsetail-splay in domain C (Figs. 3, 11, and 12 in the current manuscript). Nevertheless, although such a scenario could explain locally enhanced partial melting, it is unlikely to have evoked the formation of vast diatexite complexes as observed in the study area. Pervasive melting in the absence of an aqueous fluid (i.e., hydrate-breakdown melting) is also supported by geochemical analysis of cordierite-bearing migmatites in the central part of the study area (Kalt et al., 1999).*

*Therefore, after reviewing the mineral assemblages and P/T metamorphic conditions, we prefer post-metamorphic tectonic exhumation (Scenario IV) as the most plausible scenario to explain the observed juxtaposition of different anatectic grades along very sharp boundaries in the study area. To emphasize this conclusion, we extended the related discussion part in the manuscript by the above-mentioned points.*

**Comment 3**

"I have also same remarks even about thermochronological ages interpretation. I find interesting the interpretation of regional pattern of fission track ages and I agree that different ages can reflect different post-cooling vertical movement. Nevertheless it is not obvious to ascribe a depth of closure to a zircon FT age especially when you are considering one sample only. Complex thermal histories made by long persistence on partial annealing zone can produce very different age also in close samples. For this reason more information on the discussed data (e.g. track length, thermal modeling), if available, could better support the thesis of the authors.

Following the data of Vamvaka et al. 2014, it seems that the major reason for different AFT ages is related to complex thermal histories. To be sure that regional pattern of AFT and ZFT ages reflects fault activity, a more precise discussion of thermochronological data is needed."

Response from authors:
*Thank you for allowing us to clarify our interpretation of the Apatite and Zircon fission track record from the literature. First, we would like to emphasize that discussion on the low-temperature evolution of the study area in full detail is beyond the scope of our study. We used available FT data as an additional and supporting source of information confirming the presence of a km-scale yet unknown tectonic structure along the southwestern Bohemian Massif (Cham Fault), which complements our interpretations of the exposed granite inventory and topographic lineaments. In addition, FT data provide evidence for significant post-Variscan activity phases of the Cham Fault. Below, we outline the main points that indicate the presence of a significant tectonic structure that also influences the spatial distribution of low-temperature thermochronological ages.*

*We agree that complex thermal histories may result in very different apparent FT ages, even among samples located close to each other. Indeed, complex thermal histories are suggested for most of the analyzed AFT samples (Wagner et al., 1989; Vercoutere, 1994; Hejl et al., 1997; Siebel et al., 2010; Vamvaka et al., 2014). A generally enhanced paleo-geothermal gradient is thereby assumed for the study area during the Mesozoic (Vercoutere, 1994; Vamvaka et al., 2014).*

*From track length characteristics (mixed-bimodal to positively skewed), AFT data in the Naab Mountains to the west of the Cham Fault are interpreted as post-Variscan cooling ages (ca. 270 Ma) that were partially reset during late Permian to Mesozoic subsidence and burial (up to 1000 to 1400 m) followed by Late Cretaceous to Cenozoic uplift and exhumation (Vercoutere, 1994).*

*An even more complex thermotectonic record is suggested for the Bavarian Forest in the southeastern part of the study area (Vamvaka et al., 2014). Here, thermal models indicate a first reheating during mid- to late Jurassic times (ca. 160-140 Ma), which was followed by tectonically-driven exhumation and denudation during the Early Cretaceous (ca. 140-120 Ma). After a phase of stagnation, sedimentation recurred during the Late Cretaceous (ca. 95-85 Ma), which caused reheating of marginal parts of the Bavarian Forest. The latter phase is especially depicted by the "Grub" sample, which is the only sample in the study of Vamvaka et al. (2014) that is located to the west of the newly proposed Cham Fault (c.f., Figure 10 in the manuscript). Similar to the tectonic record of the Naab Mountains (Vercoutere, 1994), the final uplift phase of the Bavarian Forest was initiated in the Late Cretaceous, probably in the course of inversion tectonics related either to the Alpine collision (e.g., Ziegler, 1987; Ziegler et al., 1995) or Africa-Iberia-Europe convergence (Kley and Voigt, 2008).*

*Hence, from this record, a complex regional thermotectonic evolution of the study area is proposed based on AFT data. Nevertheless, a significant difference in the thermal evolution between the sector to the west of the Cham Fault compared to the eastern sector is undoubtedly present, as evidenced by two pronounced age clusters at either side of the fault (Jurassic vs. Late Cretaceous, c.f., Figure 10 in the manuscript). Vamvaka et al. (2014) attributed this change to the presence of a fault zone in between the sample of Grub and the remaining samples to the east (i.e., the Cham Fault), accommodating a vertical displacement of at least 1 km. This interpretation is in line with Gebauer (1984), who proposes the presence of a significant tectonic structure close to the village of Winklarn between domains C1 and C2, based on two ZFT data points located only 2 km apart that record an apparent age gap of ca. 45 Myrs (ca. 260 Ma and 215 Ma, respectively), which is again in accordance with the newly proposed Cham Fault.*

*A possible cause for the apparent differences in timing between the western sector (ZFT: Upper Permian, AFT: Jurassic) and the eastern sector (ZFT: Lower Triassic, AFT: Upper Cretaceous) might be attributed to the northwest-southeast prograding Upper Permian to Mesozoic depositional system (e.g., Meyer, 1989; Peterek et al., 1997; Schröder et al., 1997). Based on the above-mentioned gap of ca. 45 Myrs between two ZFT data points close to the village of Winklarn, Meyer (1989) placed the basin margin during the Triassic close to this locality. From this interpretation, we conclude that the eastern margin of the Mesozoic basin most likely has been controlled by the Cham Fault, which, in turn, resulted in a larger sedimentary cover and the partial resetting especially of AFT ages towards the west of the Cham Fault. This interpretation is in accordance with the AFT data towards the east of the Cham Fault, where most sample sites are thought to have been covered only with an insignificant or, in parts, even completely absent sedimentary cover (Vamvaka et al., 2014). In fact, even the formation of syn-tectonic Permian basins along the northwestern segments of the Pfahl and Danube shear zones (Schröder, 1988; Peterek et al., 1996) might have been guided by the Cham Fault.*

*In summary, although the present FT ages are complex in their nature, the very distinct spatial distribution of both ZFT and AFT ages in combination with the close proximity between the two clusters suggests that the low-temperature thermal history of the study area is essentially controlled by block motion along the Cham Fault. However, it must be noted that this block motion most likely did not contribute to the differential exposure of granite bodies at the Earth's surface during the "Cham Phase", which has probably already been completed in the Permian (Siebel et al., 2010; Hejl et al., 1997; Mielke, 1993; Galadí-Enríquez et al., 2009). To clarify this interpretation of the FT data, we significantly extended the related discussion part in the updated manuscript version.*

**Comment 4**
"The pattern of AFT ages suggests that ages get younger moving to the eastern region. It seems to suggest a correspondence between younger AFT ages and higher topography. This can suggest a process of isostatic response to the long-lasting erosion. It might be the case?"

Response from authors:
*From Figure 10 in the manuscript, it becomes clear that no discernible (and statistically reliable) correlation between altitude and age can be observed. Towards the east of the Cham Fault, samples with consistent Cretaceous AFT ages show 250–650 m differences in the present-day elevation (Vamvaka et*

*al., 2014). This is rather a result of post-Cretaceous vertical displacements (Vamvaka et al., 2014), than of a regional isostatic response of the entire area, which should result in a much better correlation between ages and elevation.*

**Comment 5**
"Why the authors speak about apparent age?"

Response from authors:
*We use the term "apparent age" to emphasize the fact that thermochronological ages may be complex in their interpretation and do not necessarily represent cooling ages. In our opinion, the term "apparent age" helps to avoid misinterpretation of FT ages, and, therefore, we would prefer to retain this term in our manuscript.*

**Comment 6**
"The authors do not touch one of the main problem regarding all the remnant Variscan massifs in Europe, and that is the persistence of high topography versus topographic rejuvenation. It could be worthy to discuss your results in the light of this debate."

Response from authors:
*Thank you very much for highlighting this special and very interesting problem. In our view, the main reason for the persistence of high topography is the repeated tectonic reactivation of tectonic structures along the nowadays exposed Variscan massifs in Europe. A major reactivation event thereby occurred during the Late Cretaceous to early Paleogene, which initiated new reverse faults and reactivated inherited tectonic structures in Central Europe and also within the study area (e.g., Meyer, 1989; Kley and Voigt, 2008). In fact, we currently prepare another manuscript on this topic. However, due to the very different mechanisms, timing, and methodological approaches necessary for investigating this younger tectonic history of the southwestern Bohemian Massif, we decided not to include this discussion point in the present manuscript. Nevertheless, the fact that the Cham Fault also forms tectonic boundaries of Cretaceous to Cenozoic geological features implies that it also played a significant role in the tectonic evolution of the southwestern Bohemian Massif during Cretaceous to Cenozoic times.*

[revised manuscript text omitted]

---

## Referee Report (RR1)

Dear Editor,

The revised paper from Andreas Eberts et al., titled "Late to post-Variscan differential exhumation and basement segmentation along the SW Bohemian Massif, Central Europe" has been improved in the formal aspect and it is suitable to be published after a minor correction. I appreciated the improvement of the thermochronological data discussion, although this part is not the main focus of this research.

Lines 510-511.

Here it could be useful to add a reference to show another example testifying the lower capability to erosion of high-grade metamorphic rock with respect to the lower-grade rock.

Lines 547-551

It is not clear how the new detected Cham fault is discussed in term of regional pattern of deformation. Add a lines on a possible interpretation of the Cham fault kinematics in term of a strike slip fault regional regime.

Paragraph about low-temperature thermochronological data has been improved undoubtedly. Nevertheless some point are still confused for me.

Lines 580-584: in the text the authors said that samples from the western block have experienced a greater sedimentary burial with respect to the sample from eastern block. I do not understand why the western AFT ages could result older than eastern ones that in turn should not be covered nor reset during Jurassic. The younger ages from eastern side suggest me a greater amount of sedimentary cover over the eastern block and subsequent reset during Jurassic and late Cretaceous (also if thermal modeling show a little reheating).

The fact that Cham fault played a role in the Mesozoic basin subsidence sounds correct. This is probably the more solid evidence of Cham effect on the AFT age distribution, considering also that authors highlight that final stage of exhumation occurred in Cenozoic for both the blocks.

Just a short comment: the correlation of AFT age and elevation has to be seen in sampling profile as vertical as possible. When sampling follow tens of km long profile, age-elevation correlation is not always expected, because of the change in topography between the center of the massif with respect to the edges. In these cases the center can record younger age at higher elevation, common pattern of slow eroding old orogens.

I am not sure in the correct use of capitalized letter for the informal chronological terms such as early, middle and late. Please check at the: https://stratigraphy.org/guide/defs

---

## Author Response (AR2)

Dear Editor,
dear Reviewer,

Thank you very much for your constructive comments and suggestions. We appreciate the re-evaluation by reviewer 2, which helped us to further improve and clarify our manuscript. Below, you will find a point-by-point reply list to anonymous reviewer 2 comments, with our responses highlighted in blue and italics. Where applicable, line numbers highlighted in red indicate text passages in the marked-up version of our manuscript that were changed according to the reviewers' suggestions.

On behalf of all authors,

Andreas Eberts

**Comments**

**Comment 1**, lines 510-511 / lines 516-519

"Here it could be useful to add a reference to show another example testifying the lower capability to erosion of high-grade metamorphic rock with respect to the lower-grade rock."

> Response from authors:
>
> *We agree that a reference could strengthen our interpretations of higher rock erodibilities being associated with higher-grade metamorphic units. However, as shown by Moosdorf et al. (2018), erodibilities can vary significantly even among metamorphic units, which precludes a general model of higher erodibilities being associated with higher-grade metamorphic rocks. In addition, as also mentioned in our manuscript, other factors like younger, post-Mesozoic differential block motion or spatial variations of joint and fracture densities have significant effects on the recent topography. We now make it more clear, however, that this observation is most likely a local phenomenon that does not necessarily depict a general trend.*

**Comment 2**, lines 547-551 / lines 553-561

"It is not clear how the new detected Cham fault is discussed in term of regional pattern of deformation. Add a lines on a possible interpretation of the Cham fault kinematics in term of a strike slip fault regional regime."

> Response from authors:
>
> *The tectonic configuration of the southwestern Bohemian Massif during the late stages of the Variscan Orogeny is interpreted as a conjugate shear system related to N to NNW directed shortening (Wallbrecher et al., 1991; Brandmayr et al., 1995; Peterek et al., 1996; Büttner, 1999; Galadí-Enríquez et al., 2010). As shown in Figure 11 in the manuscript, NNW-SSE to N-S trending tectonic structures might have been active as (I) left-lateral strike-slip faults or (II) extensional features that are oriented subparallel to the maximum principal stress. The presence of local Carboniferous-Permian outcrops located along the northwestern segments of the Danube and Pfahl shear zones thereby suggest a former extensional basin in between the Franconian Line and the Danube Shear Zone ("Naab Trough" sensu Schröder, 1988). As these outcrops are limited to the area west of the Cham Fault, we propose an important normal faulting phase of the Cham Fault during late to early post-Variscan wrench tectonics. This interpretation fits the reported ca. NNW-SSE to N-S maximum principal stress during the late stages of the Variscan Orogeny (Wallbrecher et al., 1991; Brandmayr et al., 1995; Peterek et al., 1996; Büttner, 1999; Galadí-Enríquez et al., 2010) and is in accordance with Peterek et al. (1996), who speculated on the presence of ca. NNW-SSE to N-S oriented basin-bounding normal faults close to the eastern margin of the Bodenwöhr Trough. However, as evidenced by higher metamorphic grades and extensively exposed granites to the west of the Cham Fault (i.e., domains A2 and C2), the period in between granite emplacement at ca. 320 Ma and Carboniferous-Permian basin subsidence (i.e., the Cham Phase) must have involved a significant amount of exhumation of the block to the west of the Cham Fault. Rapid uplift and erosion of the Variscan basement are thereby evidenced by granite fragments and detrital micas in sediments of the Carboniferous-Permian basins (e.g., Welzel, 1991; Mielke, 1993; Galadí-Enríquez et al., 2009).*
>
> *In general, several stress field adjustments occurred since the late Paleozoic Variscan Orogeny in the studied area (e.g., Peterek et al., 1997), which are likely to have initiated different kinematics of the Cham Fault. Repeated phases of tectonic activity with varying kinematics are also observed for most other fault zones along the southwestern Bohemian Massif (Table 2 in manuscript). We clarified the kinematics of the Cham Fault during late to early post-Variscan wrench tectonics in the manuscript and slightly adjusted Figure 11.*

**Comment 3**, lines 580-584 / lines 601-611

"in the text the authors said that samples from the western block have experienced a greater sedimentary burial with respect to the sample from eastern block. I do not understand why the western AFT ages could result older than eastern ones that in turn should not be covered nor reset during Jurassic. The younger ages from eastern side suggest me a greater amount of sedimentary cover over the eastern block and subsequent reset during Jurassic and late Cretaceous (also if thermal modeling show a little reheating).

The fact that Cham fault played a role in the Mesozoic basin subsidence sounds correct. This is probably the more solid evidence of Cham effect on the AFT age distribution, considering also that authors highlight that final stage of exhumation occurred in Cenozoic for both the blocks."

Response from authors:
*AFT ages to the west of the Cham Fault are interpreted to record post-Variscan cooling ages (ca. 270 Ma) that were only partially reset during late Permian to Mesozoic burial (up to ca. 1400 m), thus representing mixed ages (Vercoutere, 1994). In contrast, no evidence exists for a significant sedimentary cover during the Permian to Mesozoic for the southeastern Bavarian Forest, neither from thermochronological (Vamvaka et al., 2014) nor geological data (Meyer, 1989). Instead, at least until the Early Jurassic, the southeastern part of the study area is rather interpreted as a topographic high that delivered clastic input into the adjacent Mesozoic basin to the northwest (Meyer, 1989). In the Middle Jurassic, the sea might have been able to transgress also into the central part of the study area (Meyer, 1989). However, there is no evidence that a thicker sedimentary succession formed towards the east of the Cham Fault compared to the west, where sedimentation started much earlier. During the Early Cretaceous, inversion tectonics resulted in the complete erosion of the sedimentary cover in the central and southeastern parts of the study area, as evidenced by Upper Cretaceous sediments on top of the crystalline basement (e.g., at the Grub locality, Wilmsen et al., 2010; Meyer, 1989).*

*The younger AFT ages to the east of the Cham Fault could thereby be explained by enhanced exhumation due to Late Cretaceous inversion tectonics (e.g., Kley and Voigt, 2008; Voigt et al., 2021). As evidenced by the thermochronological record, thrusting of basement blocks led to km-scale exhumation of numerous basement domains in Central Europe during this time (von Eynatten et al., 2021). The generally higher elevation to the east of the Cham Fault might be a remnant of this tectonic impulse. In addition, the NW-SE increasing thicknesses of Upper Cretaceous sediments in the "Regensburg-Straubing Trough" in front of the Danube Shear Zone (Führer, 1978; Unger and Risch, 1991) imply a more intense reactivation of the southeastern segment of the Danube Shear Zone. This also supports an enhanced Late Cretaceous exhumation of the adjacent area to the north of this fault segment (i.e., domains A1, B, and C1). If we assume only an insignificant sedimentary cover during the Late Cretaceous in the study area (≪1 km), subsequent vertical displacement along the Cham Fault must have been greater than 1 km (Vamvaka et al., 2014) and was probably in the range of 1.5 to 2 km. We extended the paragraph on the FT age interpretation to clarify our interpretation.*

**Comment 4**

"the correlation of AFT age and elevation has to be seen in sampling profile as vertical as possible. When sampling follow tens of km long profile, age-elevation correlation is not always expected, because of the change in topography between the center of the massif with respect to the edges. In these cases the center can record younger age at higher elevation, common pattern of slow eroding old orogens."

Response from authors:
*We agree that a large lateral spacing between AFT samples makes it more difficult to reveal a relationship between ages and elevation. However, although lateral distances are small for data taken in the area of the Naab Mountains in domain C2, no distinct correlation between age and elevation is observed here (Vercoutere, 1994).*

**Comment 5**

"I am not sure in the correct use of capitalized letter for the informal chronological terms such as early, middle and late. Please check at the: https://stratigraphy.org/guide/defs."

Response from authors:
*In fact, the terms "early, middle, and late" are just as formal as "lower, middle, and upper", at least if used in the correct way. In contrast to "lower, middle, and upper", which are used to describe chronostratigraphic units (i.e., "time-rock units", eonothems/erathems/systems/series/stages), "early, middle, and late" are used to describe geochronological units (i.e., "time units", eons/eras/periods/epochs/ages). For more details on the correct usage of these terms, please see the work of Zalasiewicz et al. (2013). Hence, if used in the correct way, capitalization follows the same rules for both "early, middle, and late" and "lower, middle, and upper" (c.f., "Geologic Time Scale" of the GSA, Walker et al., 2013). We double-checked the correct usage of capitalized letters for the above-mentioned terms.*